# China's non-ferrous metal recycling technology convergence and driving factors: A quadratic assignment procedure analysis based on patent collaboration-based network structural hole

Kai Luo, Shutter Zor *

School of Accountancy, Wuhan Textile University, Wuhan, China

* shutter_z@outlook.com

## Abstract

Technological convergence is an important organizational innovation capability, essential for improving the core competitiveness of green and sustainable industries. However, studies have mainly focused on measuring technological convergence and have ignored the factors that affect the capabilities of such convergence capabilities. Thus, this study attempts to bridge this gap by providing an in-depth analysis of the impact of structural holes in inter-organizational technical cooperation networks. This technological convergence is studied from the perspective of a patent. It also considers the moderating effects of the degree of patent cooperation and the cooperation distance. It employs the social network theory to construct inter-organizational patent cooperation, and technological convergence networks, to facilitate the analysis of the effect of structural holes on such a convergence. It empirically examines 52 non-ferrous metal recycling organizations, with close patent cooperation. Accordingly, the structural hole constraint index by the inter-organizational patent cooperation network, shows a positive U-shaped curve relationship with technology convergence. Moreover, the degree of inter-organizational patent cooperation strengthens the positive U-shaped relationship, and the distance of cooperation weakens the influence of the structural hole constraint index on technology convergence. Therefore, managers and policymakers should encourage more industry-academia-research or patent alliances and designate policies to promote such cooperation.

## 1. Introduction

The recycling industry is a strategic emerging industry in China, which can significantly relieve the pressure on resources and the environment, thus promoting the construction of new urbanization and facilitating industrial restructuring [1]. Some scholars define the renewable resource industry as a series of activities regarding the recycling of renewable resources [2]. Non-ferrous metal recycling is an important renewable resource industry, and the market

**Data Availability Statement:** All data files are available from the Harvard Dataverse database (https://doi.org/10.7910/DVN/EO2R2R.)

**Funding:** This research was funded by National Natural Science Foundation of China (Grant Number 71902151), Chinese Ministry of Education Humanities and Social Sciences Foundation Project (Grant Number 18YJA630113) and the Humanities and Social Sciences Project of Hubei Provincial Education Department (Grant number 19Q082). The funders had no role in designing the study, in the collection, analysis, or interpretation of data, in the writing of the manuscript, or in the decision to publish the results.

**Competing interests:** The authors have declared that no competing interests exist.

demand for metal materials and products is increasing. It is challenging for individual enterprises to develop effective technological innovation to recycle copper and lead. Furthermore, the technology is dated and requires the integration of complex technology [3–5]. Effective industry-academia-research cooperation to promote such integration with green technologies, such as resource recycling technology, can effectively enhance the development trend to break through the 'neck' of key common technologies, boost innovation capability, improve product quality, and achieve industrial restructuring and upgrading [6]. It is imperative to study effective cooperation between organizations to promote technology convergence.

Forming research and development (R&D) cooperative networks between organizations is an important way to implement technological innovation in high-tech industries [7]. Acquiring heterogeneous resources from collaborative networks is a vital function of network locations [8]. Granovetter [9] showed that the activities of organizations, which should not be limited to internal activities, should be extended for effective collaboration with other innovation organizations. When the uncertainty and ambiguity of technology deepens, organizations focus more on acquiring external resources for iterative updates [10]. Proximity cooperation enabled the organization to quickly cross the structural hole, break organizational boundaries to access diversified and heterogeneous technological resources, conduct effective technology convergence, and accelerate organizational innovation. Hence, proximity technology cooperation and structural holes can bring effective technology convergence. Thus, the mechanism of influence of structural holes and patent cooperation networks on technology convergence is worth studying [11].

However, prior studies show lesser engagement with China's non-ferrous metal resource recycling technology. Furthermore, few studies explore the mechanism of structural holes' influence on technology convergence. Therefore, this paper begins by describing the trends in the convergence of the analytical techniques for co-occurrence-based technology in non-ferrous metal recycling. Technology convergence can be divided into the following categories. The first category is based on whether the types of organizations cooperating in R&D projects are identical. If they belong to the same category, the technologies used may be similar, instead of two completely different technologies being integrated together. If the types of cooperating organizations belong to different categories, it is possible that two different types of technologies are integrated together [12]. In addition, some scholars have used a measurement approach to study technology convergence, based on academic literature and input-output (I/O) tables and research data [13–15]. The shortcomings of the aforementioned measurement techniques include difficulties in obtaining the limited data, a lag in the data, the requirement of a long period of observation, and the reflection of the integration of only industries and applications, without reflecting the integration of new technologies. However, the current popular measurement of technology convergence method is based on the co-occurrence of the International Patent Classification (IPC) number, for patented data, fused with social network analysis technology and patent citations [16–18]. In this study, the co-occurrence of patent data and IPC helps in constructing a patent cooperation network, to describe technology convergence of non-ferrous metal recycling based on social network theory. The value of co-occurrence of two IPC classification numbers, in a patent network, represents the number of patents in which these two IPCs occur simultaneously. Previous studies measured the Herfindahl index [19], entropy [20], and total number convergence patents [21] based on patent data, yet could not observe the process and degree of convergence of individual technology nodes in the overall network. The absolute number of patents also could not describe the dynamics of technology convergence, and different technological areas involved in convergence. However, a patent document with two or more IPC classification numbers implies that the patent involves multiple technologies, reflecting the source and development trend of the

technologies and their applications [22]. This research constructs a patent network of IPC co-occurrence, based on the data from the patent information platform of key industries, of the State Intellectual Property Office of China. This can observe the composition of technologies, and the degree of integration of different technological nodes in the network. The patented information platform provides the information of non-ferrous metal recycling patents, primarily applied by innovative Chinese organizations. Describing the technological convergence dynamics proves beneficial for this study.

Hence, the second objective of this paper is to employ the social network theory and construct an inter-organizational R&D cooperation network, specifically divided into structural hole constraint index matrices. This analyses the effect of structural holes on technology convergence, under the contextual factors of patent cooperation degree and distance. Previous studies of R&D cooperation were based on the number of organized projects, knowledge output, technological diversity, and the number of employees involved in such R&D projects [12]. These are absolute data, not reflecting the flow of technology, the degree, and the specific organizational structure of cooperation. Faust and Wasserman [23] defines "one-mode" network, which speaks of some scholars measuring R&D cooperation with patent cooperation network. Patents are the R&D outputs that reflect the information and innovation outcomes of technology and its application, performances of collaborated R&D, and innovation paths [24]. Using the social network approach to construct a one-model patent cooperation network, this study observes the number of cooperated patents between two organizations in the network, reflecting the degree of R&D cooperation between them. The mobility of knowledge depends on the flexibility of the overall network, the nature of each nodal organization and the distance between them. However, this one-model social network cannot analyze the relationship between each network. For example, it cannot portray whether each nodal organization in the patent cooperation network plays the role of an intermediary or a core collaborator. It also cannot portray whether a structural hole (bridge) in the network, has any effect on the performance of innovation. At the same time, the advent of the era of big data and the development of the network integration processes have complicated the network formed by the relationship between individuals and groups. The general multiple regression analysis method cannot explore the non-independent relationships between different networks, and the covariance problem makes the ordinary least squares (OLS) method, based on time series and panel data, invalid. In this study, we try to use the quadratic assignment procedure (QAP) test to examine the hypothesis of the "relationship-relationship," and analyze the relationship data of the organization's patent cooperation. This method is based on the QAP test, which helps examine the relationship data under the organizational patent partnership and other factors affecting technological convergence, solving the auto-correlation problem, and producing relatively unbiased statistical results [25–27].

This reminder of the study is structured as follows. Section 2 presents the theoretical basis and research hypothesis. Section 3 presents the research design. Section 4 illustrates the results. Section 5 discusses the results and concludes the study. Fig 1 illustrates the theoretical framework model for this study.

## 2. Theoretical basis and research hypothesis

### 2.1. Structural hole and technology convergence

Technological convergence is a significant overlap of certain industries in two different technological fields, and technological convergence is inseparable from industrialization development [28]. It was not until after the 1980s that it gradually garnered scholarly attention, where studies noted that the boundaries of at least two disjointed technological fields started to blur

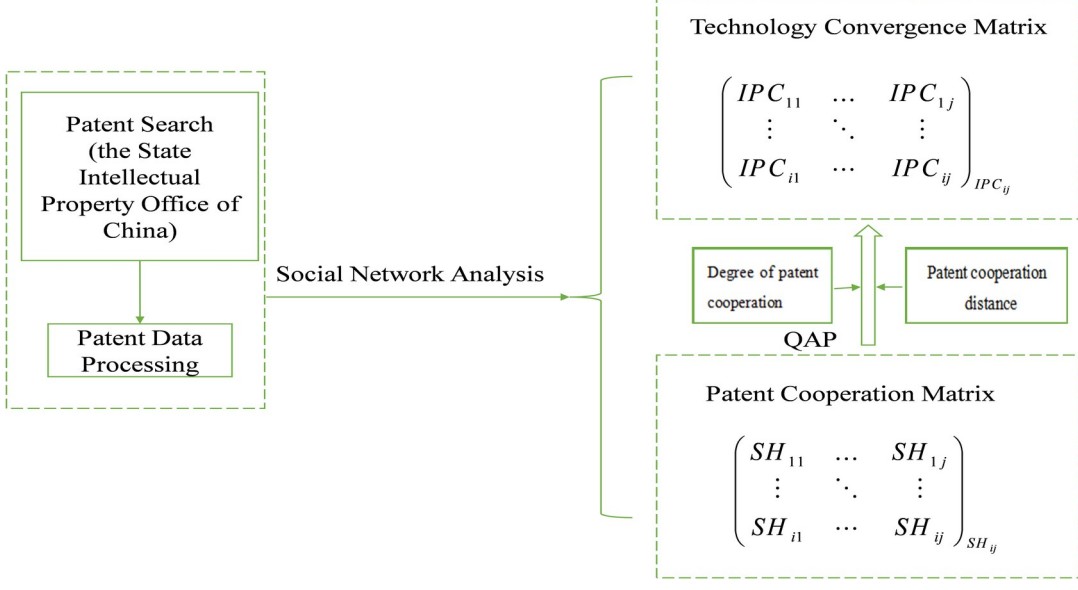

**Fig 1. The framework of research.**

and gradually disappear, birthing new technological and industrial fields, the essential process of which is the transfer of different technological fields and the shared formation of a new technological unit [17, 25, 29]. Some scholars [17, 22, 30] propose a method based on patent co-occurrence analysis to measure cross-industry convergence and employ a patent co-occurrence network to measure technology convergence in the top 30 Korean manufacturing firms and Korean patents in security technology. They suggest ways to identify core technologies, positing that technology convergence facilitates open innovation.

According to the characteristics of technology convergence, an organization must acquire external and internal knowledge to digest, absorb, and converge into new technologies. Structural holes bridge the acquisition of external knowledge technologies, and the more structural holes organizations occupy in the cooperative network, the more enterprises often intermediate among organizations with which they are associated, improving the efficiency of knowledge spill-over and the ability to control information [31, 32]. Such organizations are usually favorable in information, control, and independent R&D. Organizations occupying structural holes can access heterogeneous, non-redundant, and timely knowledge and information from both ends of the structural hole, thus facilitating the recombination of this knowledge and technology into a new generation [33]. Nodes in the intermediary bridge position have the advantage of independent R&D, and structural holes are less constrained and less susceptible to behavioral norms and paradigms of thinking from other organizations, which facilitates the development of organizational innovation [33, 34]. Moreover, technology convergence is the reorganization between different processes, industries, and technologies to produce new processes and technologies [35]. Organizations that bridge structural holes can filter out redundant information, retain non-redundant information, and are unconstrained by other organizations. They may engage in innovative activities with lower costs and greater independence than other organizations, subject to interference, such as intellectual property risks in searching and absorbing external knowledge technologies. The control advantage allows organizations to gain greater bargaining power and control over resources or outcomes by bridging connections in R&D collaborations, effectively building strategic alliances and inducing

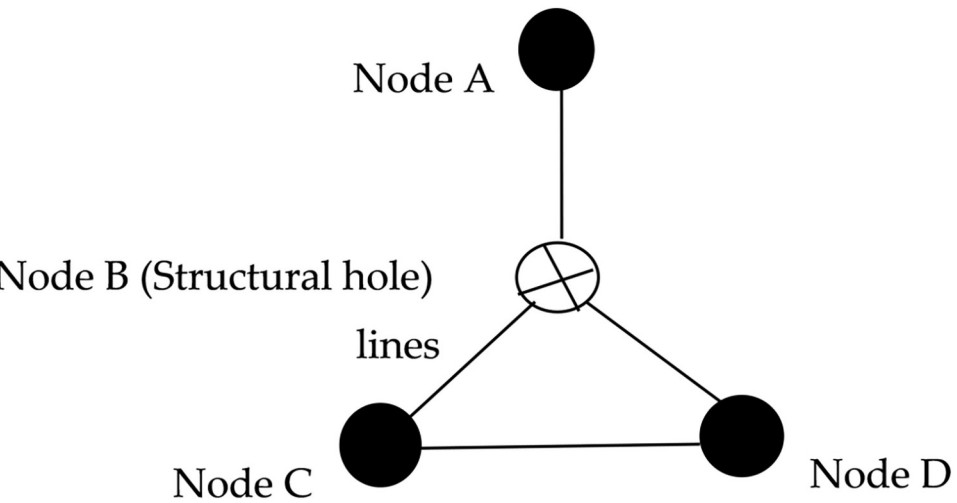

**Fig 2. Structural hole creation.**

favorable decisions [34, 36, 37]. Thus, organizations with fewer structural hole constraints can leap across the structural holes in the collaborative network by leveraging their ability to control external resources to facilitate technology transfer, absorption, and digestion and promote technology convergence efficiency. Fig 2 illustrates the structural hole characteristics.

Nevertheless, excessive non-redundant relationships can add a cost burden to organizations unconstrained by structural holes, including the costs of searching for access to non-redundant information and processing, assimilating, integrating knowledge [38]. Thus, being unconstrained by a structural hole hinders the uptake of different knowledge by nodal organizations, negatively impacting technology convergence. Hence, the structural hole advantage is likely to have its transition points, and independent organizations unconstrained by the structural hole can hinder the incremental innovation development process of other organizations, negatively affecting their innovation capabilities [39]. Further, Firms may have reduced returns to information advantage given their limited level of awareness [40]. The lack of specialization and focus makes it challenging to reorganize non-redundant knowledge in the current technological trajectory, reducing incremental innovation capabilities [41]. In technology cooperation networks, the more they rely on cooperation between different organizations to establish links, the greater the structural hole constraint, the more favorable the sharing of knowledge and technology, the more common technologies can be formed, and the more favorable the technology convergence. Therefore, we propose the following hypothesis:

**Hypothesis 1 (H1).** *There is a positive U-curve relationship between structural hole constraints and technology convergence.*

## 2.2. The effect of moderation between structural holes and degree of patent cooperation on technology convergence

The social network relationship via patent cooperation makes it easier for the nodes in the network to obtain social capital, which can bring a competitive advantage to the enterprise [18, 42, 43]. The study of social capital has gradually shifted from the study of "individuals" to "relationships," and network relationships have become an important direction for the study of social capital [37]. Organizations are increasingly embedded in networks with customers, suppliers, universities, and other institutions, blurring organizational boundaries [44]. Studies

have shown that firms' acquisition of competitive advantage hinges on the network relationships that acquire, allocate, and exchange these resources [45], given that network relationships are inimitable social capital in accessing unique resources and capabilities. Network relationships directly influence the strategies of players in a competitive field [46]. In contrast to traditional social capital theory, which relies on "human connections," some innovative agents establish network relationships through patent cooperation.

Open innovation theory suggests that organizations gradually move from independent to collaborative innovation [47]. Only after the common technology is successfully developed and solved can other technologies be rapidly released by promoting new products, processes, and technologies. Through patent cooperation, both parties can effectively grasp the common technologies they possess and conduct technology crossover, which is conducive to technology convergence. However, an inter-organizational patent cooperation network can acquire heterogeneous resources that are conducive to complementary knowledge. For example, in the smartphone field, Huawei and Ericsson continue to sign global patent cross-licensing agreements and cooperate with different organizations, such as universities and research institutions, to apply for patents. The degree of patent cooperation network indicates the number of patents jointly applied for between organizations in the patent cooperation network. Moreover, the higher the number of joint patent applications, the more frequently organizations connect, and the higher the degree of cooperation [48]. A higher network density constrains organizations and is inconducive to new knowledge innovation across structural holes [11]. However, in a dense network, inter-organizational ties increase. Frequent exchanges also generate redundant connections that are detrimental to acquiring unique knowledge, and the cost to filter redundant information increases [41, 49]. Thus, non-redundant contacts of the organization first decrease in a proprietary collaborative network, after which knowledge homogenization increases. The cohesive nature of collaboration density reduces the complexity of innovation validation, which ensures that organizations unconstrained by structural holes have access to more appropriate information resources for their current technology trajectory in a short period. However, with frequent patent cooperation in a dense network, inter-organizational exchange and cooperation induce a greater degree of dependency constraint, deeper mutual communication and understanding between organizations, knowledge and information sharing, and mutual absorption and integration, which reduces the cost of searching for non-redundant information and the risk encountered by organizations, thus promoting technological convergence. Therefore, we propose the following hypothesis:

**Hypothesis 2 (H2).** *The degree of patent cooperation has a positive moderating effect on the positive U-shaped relationship between structural hole constraints and technology convergence.*

## 2.3. The effect of structural hole and patent cooperation distance interaction on technology convergence

Closer organizational proximity, better knowledge mobility, and high efficiency can quickly compensate for the lack of an organization's innovation resources and promote corporate innovation activities. Thus, the organization need not waste time, human, and financial resources to find heterogeneous resources across regions [33, 50]. The closer the distance between organizations, the faster is the exchange of information since information access to cross-regional technologies is easily formed [51]. Moreover, the closer the organizations in a cooperative network, the easier it is to cross more structural holes, occupy rich structural

capital without external information constraints, and cooperate effectively [26, 52–54]. Therefore, we propose the following hypothesis:

**Hypothesis 3 (H3).** *There is a negative impact of regulation of structural hole constraints on technology convergence by patent cooperation distance.*

# 3. Research design

## 3.1. The variables of network matrix construction

**3.1.1. Dependent variable.** Researchers generally accept patent data as a measure of technological convergence [20]. Prior studies [16, 22, 28] considered the presence of technological convergence by the presence of at least two different IPC classification numbers in the patent literature and counted the number of co-occurring IPCs appearing in different patents to construct an IPC co-occurrence matrix to measure the degree of technological convergence, with co-occurring IPC classifications to four bits. In this study, the IPC classification number is accurate to seven digits to measure technology convergence more accurately and carefully. Moreover, we count the frequency of association between two IPC classification numbers in the patent literature, as shown in Fig 3.

In this paper, we analyze the degree of technology convergence of networks and nodes as a whole based on the construction of IPC co-occurrence networks using Ucinet. The degree of technology convergence is judged based on the similarity of technologies between different technology domains, and the higher technology similarity between two technology nodes represents the greater degree of convergence. The degree of technological similarity is measured by the Jaccard index of two nodes in the network [55]. The Jaccard index can transform the correlation matrix into a correlation coefficient matrix that expresses the strength of association and similarity between the departments; it is calculated as follows:

$$J_{ij} = C_{ij}/(C_i + C_j - C_{ij}), \tag{1}$$

where $C_i$ and $C_j$ denote the frequency of appearance of ministries $i$ and $j$, respectively, measured by the number of patents filed by each, and $C_{ij}$ denotes the frequency of association between ministries $i$ and $j$, measured by the number of patents filed in cooperation. The index can make two closely (distantly) related ministries appear even closer (more distant), analyze the relationship and structure between technologies, and study technological convergence.

**3.1.2. Independent variable.** Given the construction of a patent cooperation matrix from the joint patenting of two organizations, the structural hole is measured using the structural

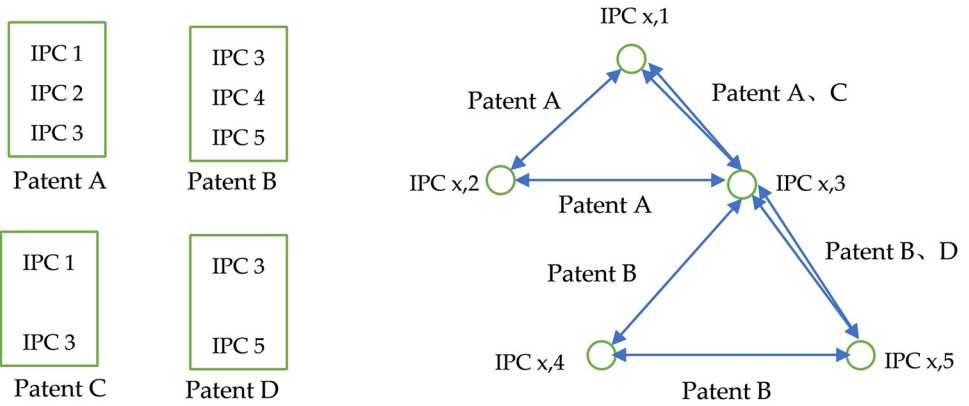

**Fig 3. Technology convergence network.**

hole constraint index [11, 44]. The structure hole is a key path for obtaining heterogeneous, novel knowledge, and its intermediary function reduces the cost of knowledge search and avoids the high cost of maintaining strong relationships [56], as demonstrated in Eq (2):

$$SH_{ij} = \left(p_{ij} + \sum_q p_{iq}p_{qj}\right)^2 \tag{2}$$

$P_{ij}$ reflects the intensity of the direct relational input of node $i$ to node $j$. It represents the proportion of direct and indirect investments of node $i$ in connection with node $j$, reflecting the relational constraint of node $i$. $P_{iq}$ represents the proportion of the total number of relationships in which actor $i$ is invested in q out of the total number of relationships, and $P_{qj}$ represents the proportion of the total number of relationships in which actor q is invested in j out of the total number of relationships, and q represents the intermediary or bridge role in network relationships in the network. $p_{ij}$ is equal to the multiplication of $P_{iq}$ and $P_{qj}$. The constraint index ($SH_{ij}$) is used to portray the advantages of node $i$ across the structural hole. The smaller the constraint index, the stronger the inter-mediation ability of the node, the less redundant the resources, and the richer the structural capital.

**3.1.3. Moderating variables and control variables.**   Patent collaboration refers to two and more patent owners working together as patent applicants [18, 30]. Patent cooperation is an important form of organizational cooperation that reflects the implementers of technological convergence and sources of innovation. Based on the social network analysis method, the patent cooperation network matrix $A = [a_{ij}]$ $(i = 1, 2,... n, j = 1, 2,... n)$. If there are joint patent applications between $i$ and $j$, then $a_{ij}$ = number of joint patent applications. Otherwise, $a_{ij} = 0$.

Based on the social network approach and the cooperation distance, this study employs the method of setting dummy variables to construct the cooperation distance matrix, where the value is $1$ if two organizations are in the same province; otherwise, it is $0$. The closer the cooperation distance, the more nodes in the cooperation network cross the structural hole, the lower the cost of acquiring knowledge information, and the less the cooperation is constrained by the structural hole. Moreover, the more the cooperation benefits technology transfer and flow, the more different-technology convergences can occur.

This study employs the nature of the organization and the sum of the accumulated patent stock between organizations as control variables. The nature of the organization is set as a dummy variable and is given the value $1$ if the two organizations are different types of organizations in the patent cooperative network relationship; otherwise, it is $0$. Further, we employ the cumulative sum of the patent stock between two organizations to denote the total sum of the patent stock of the two organizations.

## 3.2. Research methods and model

In order to analyze the degree of technological convergence among non-ferrous metal recycling technologies, we selected the main 52 IPC classifications involved in these technologies. Patent applications containing these 52 IPC classification numbers were obtained from the Patent Information Service Platform for Key Industries of the State Intellectual Property Office of China, spanning the period of 1985–2019. This platform has a dedicated non-ferrous metal recycling technology patent database, which contains all information about the relevant patented documents filed in China, such as IPC classification, year of filing, number of patents under each sub-IPC classification, major patent applicants and co-innovators, and number of co-filed patents. These patents help us observe the trend of the number of these patents, that of the associated IPC patents, and the main co-innovators filing patents.

The construction of the IPC co-occurrence matrix $J_{ij}$ is based on the patented technologies associated with these 52 IPC classification numbers. In this matrix, the vertical and horizontal axes are the fields of non-ferrous metal recycling technology subdivision, and the two technological field elements are the number of fused patents containing the two common IPC classification numbers. The Jaccard index of the two nodes in the matrix is calculated according to Eq (1), and the overall technological convergence matrix is constructed. In order to analyze the overall network and the convergence trend of each technology area in the network, we analyze the normalized degree, closeness, and betweenness of the nodes in the network based on the social network approach to measure the role of each technology area in the convergence network and visualize the $J_{ij}$. matrix using Gephi software to make the technology convergence more intuitive. The normalized degree of a node is the number of nodes directly connected to that node. The evolution of the network structure is explained by building a model of technology convergence network formation. The degree factor, as a key factor affecting the evolution of the network, reflects the core and key technologies of fusion [25]. Closeness can determine the distance of nodes in the fusion network to disseminate information [56, 57], and the larger the value, the shorter the distance from the node corresponding to that value to other nodes, indicating better mobility and faster fusion of technologies. Betweenness indicates the medium ability to transfer knowledge, which can be described as the potential influence of a technology, and if a technology has high intermediary centrality, it can be defined as a medium with high potential to facilitate knowledge transfer in different domains. In addition, to further analyze this technology convergence, we perform hierarchical cluster analysis of network nodes using CONCOR, a hierarchical cluster analysis method in social network analysis. Hierarchical cluster analysis of the nodes in the network can observe the local convergence trends of the technology domains in the network, and can reveal the degree of convergence and the relative position and role of each subdivision of the technology after CONCOR clustering when the network is partitioned into modules [25].

In order to explore the structural hole based on inter-organizational patent cooperation network on $J_{ij}$. technology convergence matrix influence effect, we used QAP regression with $J_{ij}$. as the dependent variable. The main 52 cooperative patent application organizations were extracted from the above-mentioned patent literature collected in the patent database of non-ferrous metal recycling technology of the patent information service platform of the State Intellectual Property Office of China, and the patent cooperation matrix $PC_{ij}$ was constructed using the social network analysis method, which is a one mode matrix with the horizontal and vertical axis nodes represent organizations. The larger the value, the closer the degree of cooperation between the organizations, the more resources they obtain from each other, and the closer the overall network. Based on the construction of $PC_{ij}$, the structural hole constraint index between organizations is calculated based on Eq (2), as to construct the structural hole constraint index matrix $SH_{ij}$ of organizational patent cooperation as the independent variable to study the influence effect of structural hole on technology convergence. In addition, in order to study the changes of structural hole influence on technology convergence under the contextual factors of patent cooperation, this paper takes $PC_{ij}$ as a matrix to measure the degree of patent cooperation and transforms the inter-organizational values of vertical and horizontal axes in the network into cooperation distance values, which are *1* if two organizations are in the same province, and *0* otherwise, to construct the distance matrix $GD_{ij}$ of patent cooperation. We take $PC_{ij}$ and $GD_{ij}$ as two contextual factors of patent cooperation, and the above $SH_{ij}$ and $J_{ij}$ matrices are subjected to QAP regression analysis and the interaction matrices are constructed by combining $SH_{ij}$ and $SH_{ij}$ quadratic terms with $PC_{ij}$ and $GD_{ij}$, respectively, to analyze the moderating effects of cooperation degree and cooperation distance of patent cooperation. Since the above variables are network binary data, not panel and time series data,

it is difficult to use OLS regression and each variable must be independent and positively distributed. However, the nodes in the network data are interrelated with each other and have potential indirect or direct dependencies. Therefore, the assumptions of OLS regression are not satisfied. Therefore, QAP regression is used instead in this paper. This regression method uses non-parametric alignment, and in QAP analysis, the correlation coefficients of the independent and dependent variable matrices are derived after multiple rounds of serial transformations and iterations of the vertical and horizontal axes in the network, and a test statistic is obtained to test the original hypothesis of the regression equation and whether the significance level rejects the original hypothesis. When the degree of auto-correlation between variables is high, the use of QAP has a smaller percentage of errors than the OLS [25]. QAP is a research method for analyzing the relationship between each co-occurrence matrix. The study employs the QAP for regression analysis based on the analysis of technological convergence for several reasons. It solves the problem of auto-correlation between variables, allows for a method of comparing the similarity of each element in two matrices, gives the correlation coefficient between the two matrices, performs a non-parametric test, and generates relatively unbiased statistical results applicable to this study [27, 58]. The regression equations in this paper are shown in Eq (3).

$$J_{ij} = f(SH_{ij}, (SH\ square)_{ij}, PC_{ij}, GD_{ij}, PS_{ij}, OS_{ij}). \tag{3}$$

In Eq (3), $J_{ij}$ represents the technological convergence matrix, $SH_{ij}$ represents the structural hole constraint index matrix, $PC_{ij}$ represents the patent cooperation matrix, $GD_{ij}$ is the cooperation distance matrix, $PS_{ij}$ represents the patent stock matrix, and $OS_{ij}$ represents the organizational types matrix. Eq (3) follows the method for QAP regression, with the same meaning represented through the variables. Based on the results of the QAP regression, we can test the first to third hypotheses, on the grounds of coefficients and significance levels of their independent variables. Table 1 defines these variables.

## 4. Empirical results and analysis

### 4.1. Data

Data were obtained from the database promulgated by the Patent Information Service Platform for Key Industries of the State Intellectual Property Office of China (http://chinaip.cnipa.

**Table 1. Variable descriptions.**

| Variables | Symbols | Description |
|---|---|---|
| Dependent variable | $J_{ij}$ | The Jaccard index measures the degree of node association in the network and measures technology convergence |
| Independent variable | $SH_{ij}$ | Structural hole constraint index: the smaller the index, the less the node is constrained, indicating that the node is richer in structural holes and richer in heterogeneous resource acquisition, which measures the constraint degree of structural holes |
| Moderating variables | $PC_{ij}$ | The degree of patent cooperation network refers to the cooperative relationship network formed by two or more patentees who jointly act as the applicants of a patent |
| | $GD_{ij}$ | In the patent cooperation network, the cooperation distance is measured by whether the organizations are in the same province: 1 if yes and 0 otherwise |
| Control variables | $PS_{ij}$ | For the 1985–2019 period, the accumulated patents of two nodes in the patent cooperation network |
| | $OS_{ij}$ | In the patent cooperation network, if the two organizations are different types, the value is 1 and 0 otherwise |

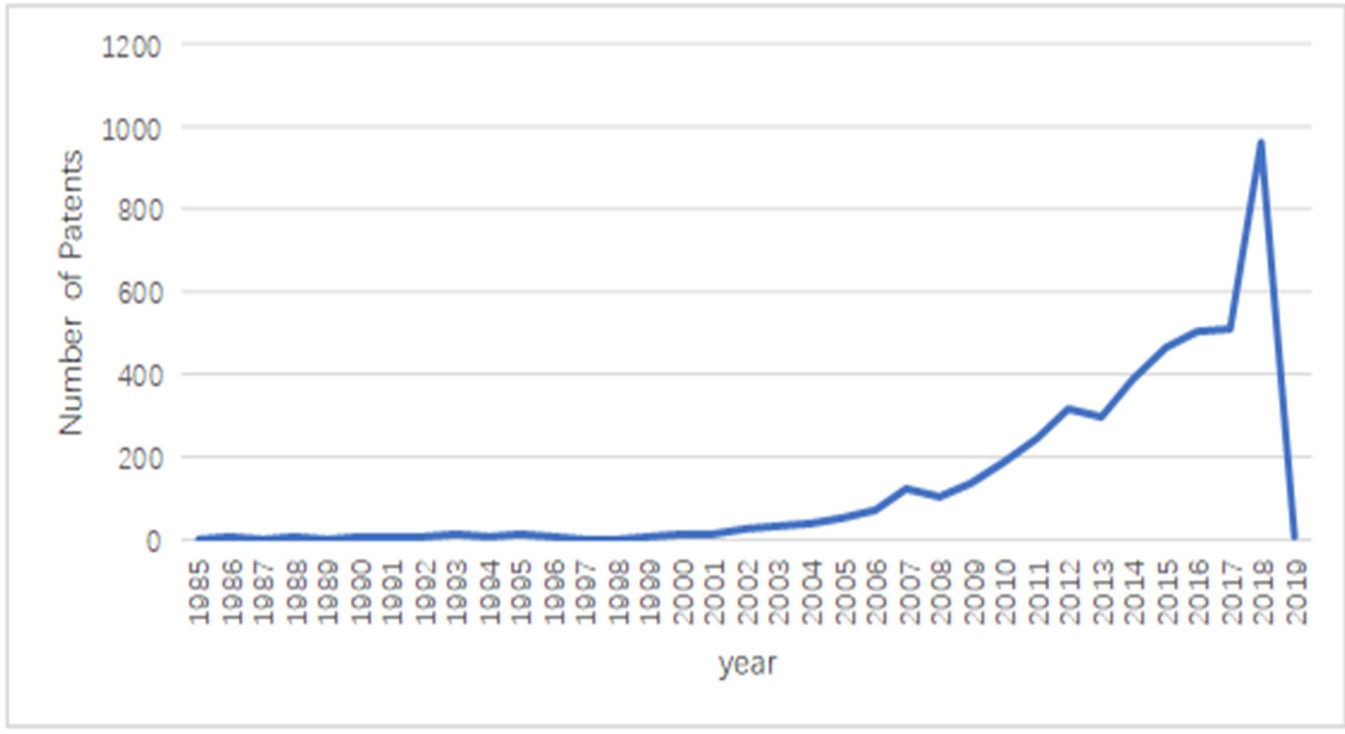

**Fig 4. Patent applications by year.**

gov.cn), it contains the main industry patent data for key analysis. The non-ferrous metal resource recycling technology belongs to the classification of non-ferrous metal smelting. Since invention patents are more convincing to innovation than designs, utility models are more convincing to innovation. Therefore, the study employs invention patent data. The search spanned the 1985–2019 period, ending in December 2019, where 4,544 cumulative invention patent applications were retrieved (Fig 4).

The Chinese patent system has a review cycle of two to three years; therefore, 2019 observed a downward trend in invention patents. China's technology patents regarding non-ferrous metal resource recycling have gone through three stages. First, in the 1985–2001 period, the annual number of patent applications did not exceed 10. China's patent protection awareness was not strong, the system was in its infancy, and the maturity of technological innovation was insufficient. Second, in the 2002–2006 period, given the continuous progress of technological innovation, the number of patent applications increased, and there was more room for technological development. Third, from 2006 to the present, given the increasing technology convergence, the patent system has improved, and the state has successively issued policies to support innovation; the trend of the increase in patent quantity accords with the development of the renewable resources industry, with more patents expected in the future. Using the text mining method, 52 organizations jointly applied for patents (including universities, research institutions, and enterprises), some of which are illustrated in Fig 5.

We employ the number of patents filed jointly by two organizations as the strength of the cooperation relationship between nodes and import the data into Excel 2010 to construct a 52×52 patent cooperation co-occurrence matrix ($PC_{ij}$) to measure the structural hole constraint index matrix. We then count the 52 IPC classification numbers to seven digits and use the number of patents associated with the two IPC categories as the linkage to construct a

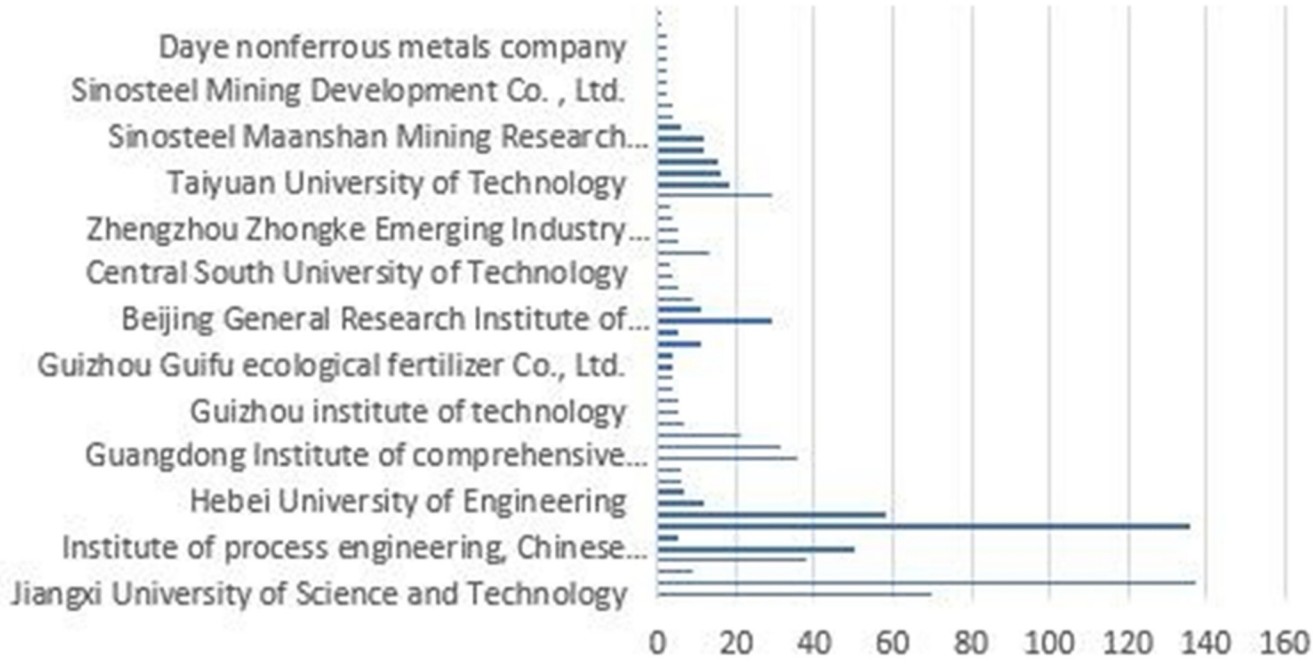

**Fig 5. Patent accumulation by major applicants (in part).**

52×52 IPC co-occurrence matrix, which we also import into Excel to measure the technology convergence matrix.

This study divides the analysis results into the following sections: technology convergence analysis, including degree, near, and intermediate centralities and iterative correlation convergence method (CONCOR) analysis; and QAP analysis.

## 4.2. Patent cooperation network and technology convergence analysis

We used Ucinet software to calculate the normalized degree of each organization node in the constructed patent cooperation matrix $PC_{ij}$ and the ranking based on the normalized degree value. Patent applications by universities dominate that of enterprises in technology innovation regarding non-ferrous metal resources recycling in China. Hence, there is room for improvement in innovation and transformation applications. Table 2 highlights the specific classification of each organization. According to the analysis results of the normalized degree, "Jiangxi University of Science and Technology," "Central South University, "and Western Mining Co., Ltd." ranked in the top three, with "Jiangxi University of Science and Technology" having the highest normalized degree value (3.922). This indicates that in the patent cooperation network, "Jiangxi University of Science and Technology" has the most connected nodes, the greatest degree of cooperation with other organizations, and the strongest technological innovation ability, and is in the core position in the network. Otherwise, from Table 2, there are 10 universities, 13 research institutions, and 29 enterprises; although universities were least represented in the sample, they have a larger normalized degree and are crucial in the cooperation

**Table 2. Classification of major patent cooperative organizations.**

| Types of Organizations | Name | Normalized Degree | Rank |
|---|---|---|---|
| Colleges and Universities (10) | Jiangxi University of Science and Technology | 3.922 | 1 |
| | Central South University | 0.218 | 2 |
| | Kunming University of Science and Technology | 0.436 | 7 |
| | Hebei University of Engineering | 2.397 | 9 |
| | Wuhan Institute of Technology | 1.089 | 14 |
| | Guizhou Institute of Technology | 1.743 | 17 |
| | Central South University | 1.089 | 29 |
| | Guangxi University | 0.218 | 36 |
| | Taiyuan University of Technology | 0.871 | 37 |
| | University of Science and Technology Liaoning | 0.218 | 38 |
| Research Institutes (13) | Hunan Institute of Non-Ferrous Metals | 1.307 | 4 |
| | Institute of Process Engineering, Chinese Academy of Sciences | 1.089 | 5 |
| | Guangdong Institute of Comprehensive Utilization of Resources | 0.218 | 13 |
| | China Coal Geological Engineering Co., Ltd. Beijing Institute of Hydraulic Engineering and Environmental Geology | 1.743 | 19 |
| | Beijing General Research Institute of Mining and Metallurgy | 0.218 | 25 |
| | Guangzhou Institute of Geochemistry, Chinese Academy of Sciences | 1.307 | 27 |
| | Changchun Gold Research Institute | 1.089 | 31 |
| | China Gold Group Corporation Technology Center | 1.089 | 32 |
| | Zhengzhou Zhongke Emerging Industry Technology Research Institute | 1.089 | 33 |
| | Shenyang Aluminium Magnesium Design & Research Institute | 0.871 | 39 |
| | Jilin Provincial Metallurgical Research Institute | 0.654 | 50 |
| | Zhengzhou Light Metal Research Institute | 0.871 | 51 |
| | China Non-ferrous Metals Technology Development Exchange Center | 0.871 | 52 |
| Enterprises (29) | Western Mining Co., Ltd. | 2.832 | 3 |
| | Western Mining Group Technology Development Co., Ltd. | 2.397 | 6 |
| | Zijin Mining Group Co., Ltd. | 1.961 | 8 |
| | Daye Non-Ferrous Design and Research Institute Co., Ltd. | 1.961 | 10 |
| | Shenzhen Zhongjin Lingnan Non-ferrous Metals Co., Ltd. | 1.307 | 11 |
| | Tianjin Shunneng Shijia Environmental Protection Technology Co., Ltd. | 2.179 | 12 |
| | Xiamen Zijin Mining and Smelting Technology Co., Ltd. | 1.961 | 15 |
| | Jilin Haorong Technology Development Co., Ltd. | 1.743 | 16 |
| | Daye Non-ferrous Metals Co., Ltd. | 1.743 | 18 |
| | Great Wall Aluminum Company of China | 1.089 | 20 |
| | Guizhou Guifu Ecological Fertilizer Co., Ltd. | 1.743 | 21 |
| | Guizhou Kexin Chemical and Metallurgical Co., Ltd. | 1.743 | 22 |
| | Jilin Gene Nickel Co., Ltd. | 1.525 | 23 |
| | Beijing Building Materials Science Research Institute Co., Ltd. | 0.654 | 24 |
| | China Ruilin Engineering Technology Co., Ltd. | 1.307 | 26 |
| | Guangzhou Zhongke Zhengchuan Environmental Protection Technology Co., Ltd. | 1.307 | 28 |
| | Inner Mongolia Dongshengmiao Mining Co., Ltd. | 1.307 | 30 |
| | China Non-ferrous Metal Mining (Group) Co., Ltd. | 0.436 | 34 |
| | Daye Non-ferrous Metals Group Holding Co., Ltd. | 1.089 | 35 |
| | Hunan Shizhuyuan Non-ferrous Metal Co., Ltd. | 0.871 | 40 |
| | Sinosteel Maanshan Mining Research Institute Co., Ltd. | 0.871 | 41 |
| | Hubei Liuguo Chemical Co., Ltd. | 0.871 | 42 |
| | Shanxi Kaixing Red Mud Development Co., Ltd. | 0.871 | 43 |
| | Huawei National Engineering Research Center for Efficient Recycling of Metal Mineral Resources Co., Ltd. | 0.871 | 44 |
| | Sinosteel Mining Development Co., Ltd. | 0.871 | 45 |
| | Sinosteel Hunan Phoenix Mining Co., Ltd. | 0.871 | 46 |
| | Bayannur Western Copper Co., Ltd. | 0.871 | 47 |
| | Hebei Ruisuo Solid Waste Engineering Technology Research Institute Co., Ltd. | 0.436 | 48 |
| | Daye non-ferrous metals company | 0.871 | 49 |

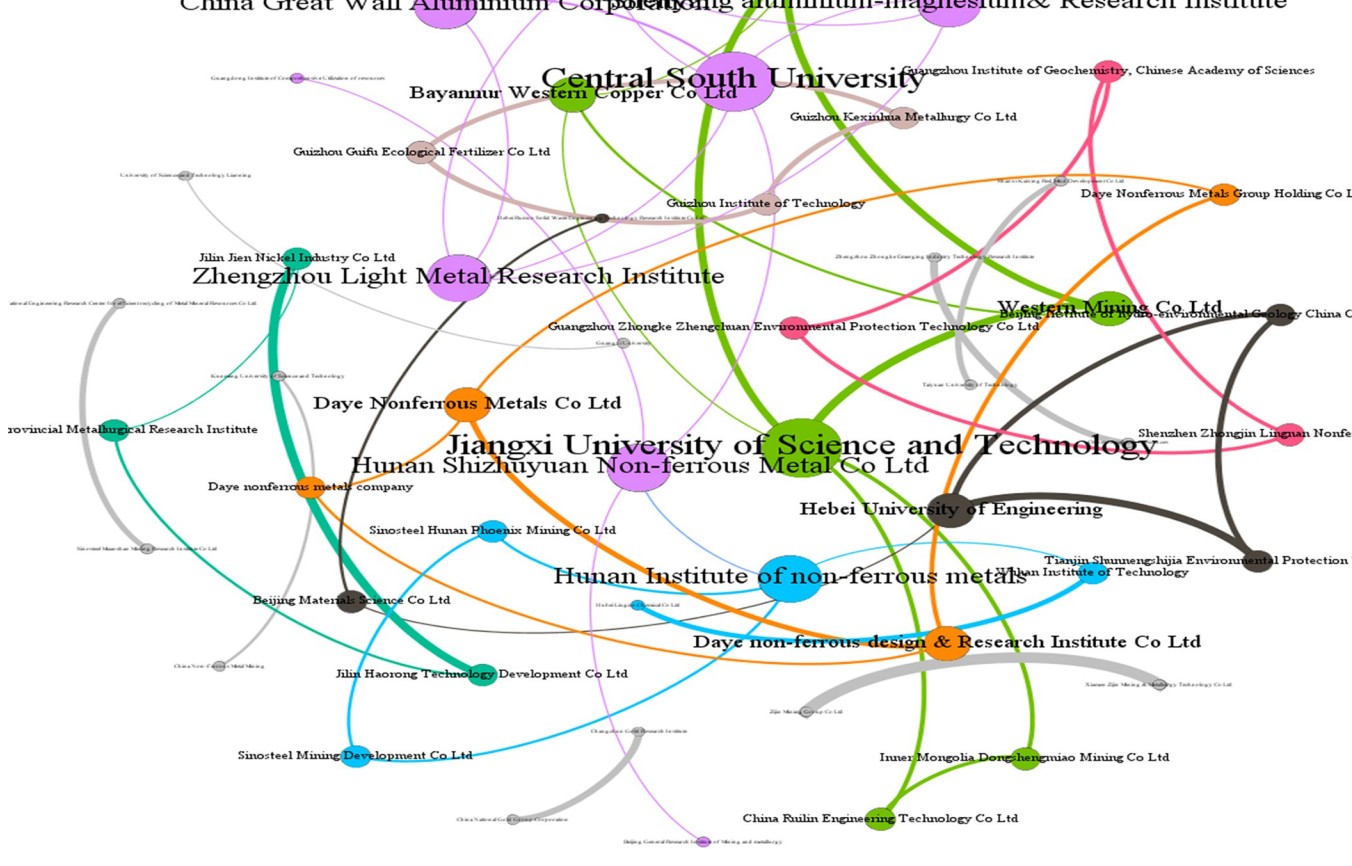

**Fig 6. Major inter-organizational patent cooperation.**

network. Universities pay more attention to open innovation and effective integration of technology. Further, visualizing the patent cooperation matrix $PC_{ij}$ with Gephi software, as shown in Fig 6, the nodes represent each organization and the co-occurring patents; the thickness of the connecting lines connecting the nodes indicate the number of co-occurring patents. In Fig 6, the size of each node indicates the degree of cooperation between the organizations, and the thickness of each path indicates the number of patents of cooperation between organizations; the higher the frequency of cooperation, the thicker the line of the path. Research institutions cooperate in greater numbers than enterprises, although their degree is smaller. Further, institutions are more capable of independent innovation (Fig 6).

We calculated the number of patents co-occurring among IPC classification numbers from the collected patent data containing 52 IPC classification numbers, and constructed the IPC co-occurrence matrix for analyzing the technology convergence network. The values of normalized degree, closeness, and betweenness of the constructed IPC co-occurrence matrix were calculated by using Ucinet software, as shown in Table 3. B03D103/02 (10.904) is the highest, which refers to "collectors," "depressants," and "ores" in the technology convergence network. This indicates that the three technology areas "collectors," "depressants," and "ores" are the most connected nodes in the technology convergence network, the most frequent fusion, and the highest degree of fusion, representing the core technology area of non-ferrous metal resource recycling. Closeness can determine the distance of nodes in the fusion network to

disseminate information, and if the distance from nodes to other nodes is shorter, indicating a better mobility and faster fusion of technologies. C22B7/00 and B03B7/00, closest to the center degree, are relatively large, constituting the overall core of the network. However, they have a smaller center degree point and a larger intermediate center degree, which shows that although the sample organization is processing scrap and other materials for production, the device and method of processing waste refuse is not the core technology. Further, it constitutes the intermediate point of the network; moreover, it can promote the integration of other technologies. Betweenness indicates the possibility of technology convergence in the future and can be described as the potential influence of a technology, and if a technology possesses high mediating centrality, a mediating node with high potential to facilitate technology convergence in different domains is considered. Table 3 shows that C22B7/00 (4.092), B03B7/00 (3.804), and C22B7/04 (3.033) have the highest betweenness values. Although their normalized degree values are not high, B03B7/00 not only has a higher betweenness value, but also a larger closeness value. This means that non-ferrous metal resource recycling in China is "working-up raw

**Table 3. IPC correlation and measurement metrics.**

| IPC class | IPC subclass | Description | Normalized degree | Closeness | Betweenness |
|---|---|---|---|---|---|
| B03D | B03D101/02 | Collectors | 17.347 | 79.688 | 2.591 |
| | B03D101/06 | Depressants | 12.465 | 75 | 1.832 |
| | B03D103/02 | Ores | 10.904 | 73.913 | 1.806 |
| | B03D1/018 | Mixtures of inorganic and organic compounds | 9.924 | 73.913 | 1.663 |
| | B03D1/00 | Flotation | 9.744 | 80.952 | 2.794 |
| | B03D101/04 | Frother | 9.804 | 76.119 | 1.93 |
| | B03D1/002 | Inorganic compounds | 6.723 | 72.857 | 0.566 |
| | B03D103/04 | Non-sulfide ores | 5.762 | 63.75 | 0.506 |
| | B03D1/02 | Inorganic compounds | 5.142 | 66.234 | 0.566 |
| | B03D1/012 | Containing sulfur | 5.122 | 68 | 0.812 |
| | B03D1/008 | Containing oxygen | 3.882 | 64.557 | 0.632 |
| | B03D1/08 | Subsequent treatment of concentrated product | 2.961 | 59.302 | 0.099 |
| | B03D1/016 | Macromolecular compounds | 2.861 | 57.955 | 0.033 |
| | B03D1/01 | Containing nitrogen | 2.021 | 66.234 | 0.946 |
| | B03D1/004 | Organic compounds | 1.801 | 56.667 | 0.051 |
| | B03D101/00 | Specified effects produced by the flotation agents | 1.140 | 57.303 | 0.079 |
| | B03D1/014 | Containing phosphorus | 1.120 | 57.303 | 0.054 |
| | B03D1/001 | Flotation agents | 0.88 | 57.955 | 0.201 |
| B03B | B03B7/00 | Combinations of wet processes or apparatus with other processes or apparatus (e.g., for dressing ores or garbage) | 9.024 | 80.952 | 3.804 |
| | B03B1/00 | Conditioning for facilitating separation by altering physical properties of the matter to be treated | 7.143 | 75 | 1.967 |
| | B03B9/00 | General arrangement of separating plant (e.g., flow sheets) | 6.323 | 73.913 | 2.035 |
| | B03B1/04 | By additives | 3.161 | 64.557 | 0.696 |
| | B03B9/06 | Specially adapted for refuse | 0.860 | 62.195 | 0.222 |

(*Continued*)

**Table 3.** (Continued)

| IPC class | IPC subclass | Description | Normalized degree | Closeness | Betweenness |
|---|---|---|---|---|---|
| C22B | C22B7/00 | Working-up raw materials other than ores (e.g., scrap, to produce non-ferrous metals or compounds thereof) | 2.821 | 75 | 4.092 |
| | C22B15/00 | Obtaining copper | 2.721 | 76.119 | 2.495 |
| | C22B1/02 | Roasting processes (C22B1/16 takes precedence) | 2.681 | 75 | 2.157 |
| | C22B3/08 | Sulfuric acid | 1.721 | 68 | 1.242 |
| | C22B7/04 | Working-up slag | 1.581 | 68 | 3.033 |
| | C22B34/12 | Obtaining titanium | 1.341 | 58.621 | 0.93 |
| | C22B1/00 | Preliminary treatment of ores or scrap | 1.16 | 67.105 | 1.913 |
| | C22B34/22 | Obtaining vanadium | 1.18 | 64.557 | 2.629 |
| | C22B26/10 | Obtaining alkali metals | 1.1 | 66.234 | 1.927 |
| | C22B3/04 | By leaching (C22B3/18 takes precedence) | 0.98 | 65.385 | 0.927 |
| | C22B59/00 | Obtaining rare earth metals | 0.96 | 60.714 | 0.926 |
| | C22B11/00 | Obtaining noble metals | 0.88 | 62.195 | 0.455 |
| | C22B19/30 | From metallic residues or scraps | 0.78 | 61.446 | 0.468 |
| | C22B3/44 | By chemical processes (C22B3/26, C22B3/42 take precedence) | 0.84 | 62.963 | 0.617 |
| | C22B23/00 | Obtaining nickel or cobalt | 0.8 | 62.195 | 0.585 |
| | C22B21/00 | Obtaining aluminum | 0.76 | 57.303 | 0.235 |
| | C22B13/00 | Obtaining lead | 0.6 | 59.302 | 0.303 |
| | C22B1/24 | Binding; Briquetting | 0.62 | 62.195 | 0.694 |
| | C22B11/08 | Cyaniding | 0.58 | 55.435 | 0.091 |
| B03C | B03C1/02 | Acting directly on the substance being separated | 1.821 | 70.833 | 2.076 |
| | B03C1/015 | By chemical treatment imparting magnetic properties to the material to be separated (e.g., roasting, reduction, and oxidation) | 1.02 | 61.446 | 0.549 |
| | B03C1/00 | Magnetic separation | 0.88 | 61.446 | 0.722 |
| | B03C1/30 | Combinations with other devices, not otherwise provided for | 0.82 | 60.714 | 0.555 |
| C21B | C21B13/00 | Making spongy iron or liquid steel by direct processes | 1.02 | 60.714 | 1.340 |
| | C21B3/06 | Treatment of liquid slag | 0.74 | 46.789 | 0.039 |
| | C21B11/00 | Making pig-iron other than in blast furnaces | 0.58 | 50 | 0.058 |
| B02C | B02C21/00 | Disintegrating plant with or without drying of the material (for grain B02C9/04) | 0.66 | 60 | 0.299 |
| C25C | C25C1/12 | Copper | 0.62 | 57.955 | 0.207 |
| C04B | C04B7/147 | Metallurgical slag | 0.6 | 50 | 0.079 |

materials other than ores (e.g., scrap, to produce non-ferrous metals or compounds thereof)," "combinations of wet processes or apparatus with other processes or apparatus (e.g., for dressing ores or garbage)," and "working up slag." They have a high potential for convergence in the areas of technology convergence networks, where they are directly or indirectly connected to other nodes, while facilitating the convergence of other technologies, and may be a key technology area for innovation in China in the future. In addition, we use Gephi software to visualize the IPC co-occurrence matrix. Fig 7 presents the network visualization. The thickness of the line represents the number of patents associated with the IPC node.

## 4.3. Concor analysis

To further analyze the technology convergence, we conducted a CONCOR calculation on the IPC association network using Ucinet. Fig 8 presents the results. The network modules are clustered into eight, and the clustering of each module indicates that the members have similar

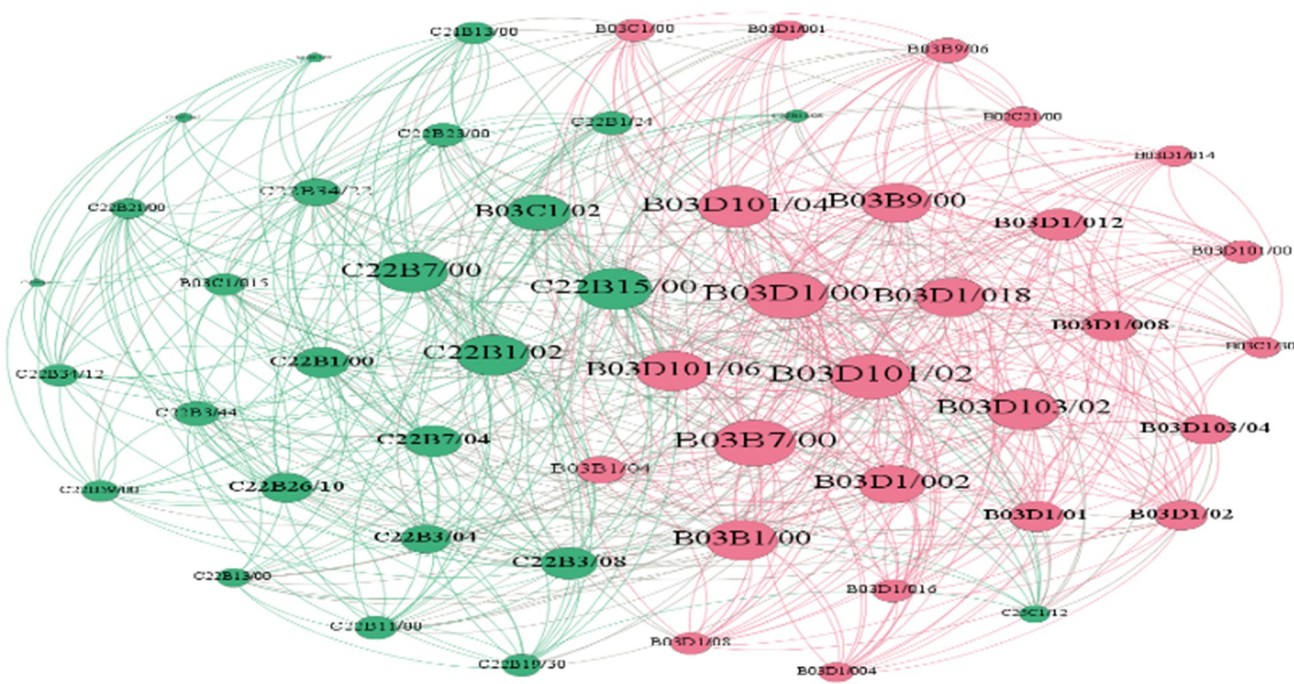

**Fig 7. IPC association network diagram.**

convergence. The first module is most closely connected to the second module and represents the maximum technological convergence, as a method for isolating useful and sustainable productive uses from waste materials. The second module is similar to the first module in that it is more convergent. The fifth module has fewer clustered nodes, but its intermediary role is obvious, as it connects the fourth module, which outputs the methodological and technological

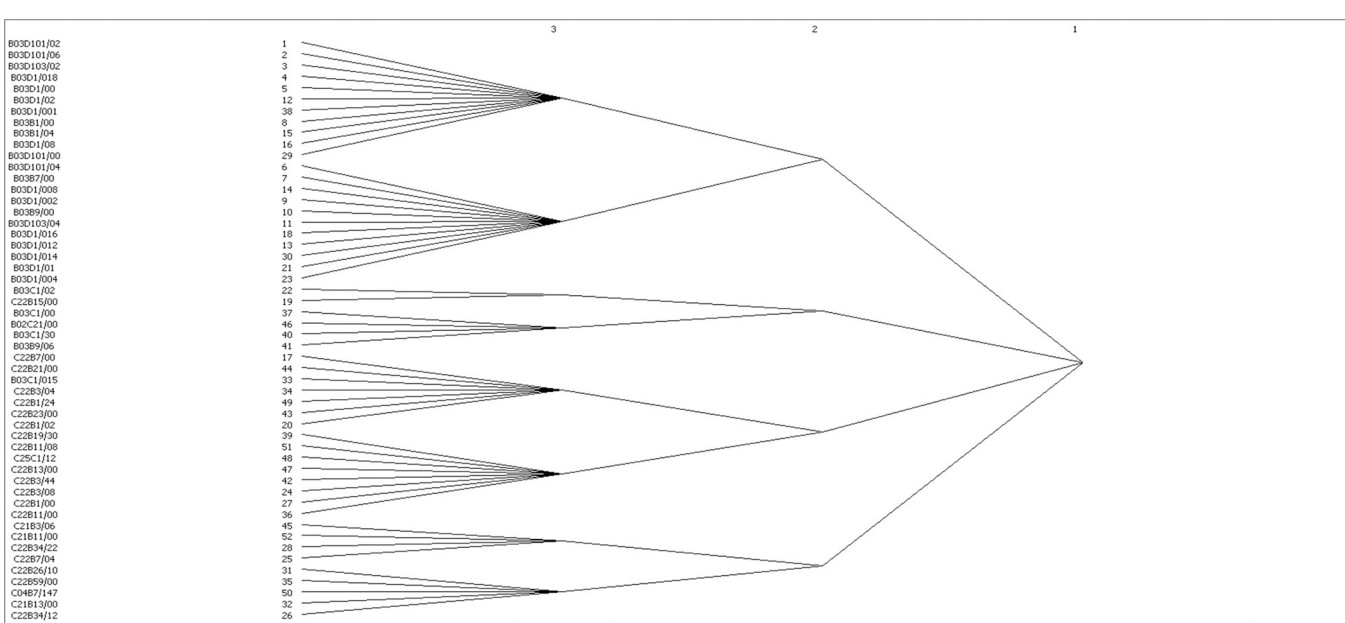

**Fig 8. CONCOR analysis.**

**Table 4. IPC correlation module density matrix.**

| Cluster | 1 | 2 | 3 | 4 | 5 | 6 | 7 | 8 |
|---------|------|------|------|------|------|------|------|------|
| 1 | 21.800 | 11.678 | 2.818 | 1.636 | 0.519 | 0.409 | 0.318 | 0.073 |
| 2 | 11.678 | 10.400 | 1.909 | 1.295 | 0.338 | 0.341 | 0.136 | 0.055 |
| 3 | 2.818 | 1.909 | 1.000 | 0.500 | 3.857 | 3.000 | 1.250 | 0.500 |
| 4 | 1.636 | 1.295 | 0.500 | 0.667 | 0.214 | 0.156 | 0.063 | 0.400 |
| 5 | 0.519 | 0.338 | 3.857 | 0.214 | 4.048 | 1.375 | 0.786 | 2.543 |
| 6 | 0.409 | 0.341 | 3.000 | 0.156 | 1.375 | 2.107 | 0.656 | 0.600 |
| 7 | 0.318 | 0.136 | 1.250 | 0.063 | 0.786 | 0.656 | 6.333 | 2.700 |
| 8 | 0.073 | 0.055 | 0.500 | 0.400 | 2.543 | 0.600 | 2.700 | 3.200 |

$R^2$ value = 0.371

devices of waste treatment to the means of separating useful materials. Moreover, it is integrated with the sixth module, the refining of non-metallic materials, which serves a certain intermediary utility. The density matrix of the eight module was calculated using Ucinet (Table 4). Table 4 shows that Modules 1 and 2 have the highest correlation density, indicating that the elements in Module 1 are most closely linked to those in Module 2 in the technology convergence network. This reflects the high degree of convergence between the technology areas involved in Module 1 and those involved in Module 2. This indicates that China has a strong technological innovation capability in the fields of flotation, selective deposition method and separation of solid materials by liquid, wind shaker or wind jigger. They have formed very mature and stable fusion technologies in these two technology areas.

## 4.4. QAP analysis

**4.4.1. Descriptive statistics.** Table 5 presents the descriptive statistics of the variable matrices. Further, to ensure the validity and consistency of the sample, the sample with fewer cooperative patents was deleted, leaving 52 organizations' invention application patents for the research data. Ucinet was used to calculate the constraint index matrix of the patent cooperation network to measure the degree of structural hole constraint.

**4.4.2. Correlation analysis.** Before conducting the QAP analysis, the correlation between each variable must be analyzed to test for potential multicollinearity between the variable data. From Table 6, it is clear that the correlation coefficients among the variables are not significant, thus indicating that multicollinearity is not a serious problem. Therefore, the constructed model is valid, and the QAP regression analysis is suitable for verifying the hypotheses proposed above.

**4.4.3. Regression results.** We use QAP to analyze the relationship between the constructed structural hole constraint index $SH_{ij}$, and Jaccard index matrix $J_{ij}$, and then explore

**Table 5. Descriptive statistics.**

| Variables | Obs. | Mean | Std. Dev. | Min | Max |
|-----------|------|------|-----------|-----|-----|
| $J_{ij}$ | 2652 | 0.027 | 0.056 | 0 | 0.667 |
| $PC_{ij}$ | 2652 | 0.109 | 0.64 | 0 | 9 |
| $PS_{ij}$ | 2652 | 34.198 | 39.291 | 1 | 273 |
| $GD_{ij}$ | 2652 | 0.058 | 0.234 | 0 | 1 |
| $OS_{ij}$ | 2652 | 0.558 | 0.497 | 0 | 1 |
| $SH_{ij}$ | 2652 | 0.003 | 0.005 | 0 | 0.034 |

**Table 6. Correlation analysis.**

| Variables | $J_{ij}$ | $PC_{ij}$ | $PS_{ij}$ | $GD_{ij}$ | $OS_{ij}$ | $SH_{ij}$ |
|---|---|---|---|---|---|---|
| $J_{ij}$ | 1 | | | | | |
| $PC_{ij}$ | 0.117*** (0.001) | 1 | | | | |
| $PS_{ij}$ | 0.205*** (0.001) | -0.022* (0.097) | 1 | | | |
| $GD_{ij}$ | 0.090*** (0.006) | 0.128*** (0.000) | -0.030 (0.197) | 1 | | |
| $OS_{ij}$ | 0.055* (0.060) | -0.007 (0.377) | 0.265*** (0.000) | -0.003 (0.438) | 1 | |
| $SH_{ij}$ | 0.045* (0.055) | 0.617*** (0.000) | 0.010 (0.223) | 0.181*** (0.000) | -0.006 (0.382) | 1 |

Notes

* $p < 0.1$

** $p < 0.05$

*** $p < 0.01$.

the influence of structural holes on technological convergence, based on patent cooperation networks. First, we use the $52 \times 52$ $J_{ij}$ matrix as the dependent variable for measuring the technology convergence network, in which the horizontal and vertical axes are permuted 2,000 times. The independent variables are $SH_{ij}$ and $(SH\ square)_{ij}$, the moderating variables are $PC_{ij}$ and $GD_{ij}$, and the control variables are $PS_{ij}$ and $OS_{ij}$. We perform QAP analysis on the above co-occurrence matrix using Ucinet software.

We estimated a total of seven models in Table 7. Specifically, Model 1 includes only control variables. Models 2 and 3 examine the direct effect of structural holes on technology convergence. We construct a quadratic matrix of the structural hole constraint index $(SH\ square)_{ij}$. Model 3 reveals a U-shaped curve relationship between the structural hole constraint index and technology convergence. Models 4, 5, and 6 show the moderating effect of the degree of patent cooperation and the distance of patent cooperation, as evidenced by the interaction effect with the quadratic terms of the structural hole constraint index and the structural hole constraint index. Model 7 is the full model with all variable matrices.

**Table 7. QAP regression results for technology convergence.**

| $J_{ij}$ | Model 1 | Model 2 | Model 3 | Model 4 | Model 5 | Model 6 | Model 7 |
|---|---|---|---|---|---|---|---|
| $SH_{ij}$ | | 0.012863 | -0.068671*** | -0.082035*** | -0.119666*** | -0.067502*** | -0.105121*** |
| $(SH\ square)_{ij}$ | | 0.086618*** | 0.068968** | 0.064904** | 0.066815** | 0.066128** | 0.068134** |
| $PC_{ij}$ | | | 0.141772*** | 0.140354*** | 0.125058*** | 0.141314*** | 0.126042*** |
| $GD_{ij}$ | | | | 0.086126*** | 0.085750*** | 0.091646*** | 0.091266*** |
| $SH_{ij} \times GD_{ij}$ | | | | | | -0.030539* | -0.030481* |
| $SH_{ij} \times PC_{ij}$ | | | | | -0.015443 | | -0.015551 |
| $(SH\ square)_{ij} \times PC_{ij}$ | | | | | 0.061336* | | 0.061256* |
| $PS_{ij}$ | 0.204813*** | 0.206259*** | 0.210093*** | 0.212847*** | 0.212935*** | 0.213801*** | 0.213886*** |
| $OS_{ij}$ | 0.000463 | 0.000816 | 0.000127 | -0.000448 | 0.000181 | -0.000581 | 0.000043 |
| Intercept | 0.000000 | 0.000000 | 0.000000 | 0.000000 | 0.000000 | 0.000000 | 0.000000 |
| N | 2652 | 2652 | 2652 | 2652 | 2652 | 2652 | 2652 |
| $R^2$ | 0.042 | 0.05 | 0.063 | 0.070 | 0.071 | 0.070 | 0.071 |
| $Adj.R^2$ | 0.042 | 0.049 | 0.061 | 0.068 | 0.068 | 0.068 | 0.068 |

Notes

* $p < 0.1$

** $p < 0.05$

*** $p<0.01$.

First, Model 1 shows the regression equations for the control variables, followed by the regression equations for the independent variables ($SH_{ij}$ and $(SH\ square)_{ij}$) (Table 7, Model 2, 3). The regression coefficients of structural hole constraint index matrix ($SH_{ij}$) and quadratic structural hole constraint index matrix (($SH\ square)_{ij}$) are negative and positive and significant, respectively ($\beta_1$ = -0.068671, p<0.01, $\beta_2$ = 0.068968, p<0.05). We found that the structural hole constraint index shows a positive U-shaped relationship with technology convergence, thus verifying H1. Hence, the more initial structural holes constrain the nodes in the network, the more unfavorable technology convergence is, and the more nodes occupy structural holes, thereby serving as intermediaries to promote the acquisition and absorption of knowledge and technology and effective technology moderation. As cooperation deepens, the organization crosses more structural holes, further increasing the risk cost, which is not as efficient as the cooperation sharing interdependence between nodes. This finding enriches the research on the impact of structural holes on innovation performance, Wen et.al argued [59] that the more structural holes that a firm crosses in an R&D network, the less constrained it is by the connected nodes, and the more complementary non-related diversified knowledge it can acquire, thus effectively absorbing and integrating internal and external knowledge and promoting the firm's exploitative and exploratory innovation outcomes. The innovation performance of the above studies is relatively homogeneous, and does not consider the linear relationship between structural holes and innovation performance. Instead, we examine the linear relationship between structural holes and technology convergence from the technology convergence perspective. We find a U-shaped relationship between the two. The less constrained an organization is in a patent cooperation network, the more technology it can absorb, the more nodes it can cross, and the more internal and external resources it can effectively access to promote technology convergence. After passing the inflection point, with the deepening of cooperation, the organizations' connection strengthens, and the ability of collaborative innovation increases. This proves that a joint cooperation can promote technology convergence better than individual enterprises' integration of knowledge and technology. At the same time, the resource-based theory found that the heterogeneous resources of organizations can help improve competitiveness. Our research presents the smaller the structural hole constraint in the patent cooperation network can absorb more external resources and integrate technologies. The results enrich the resource-based theory.

Second, the regression coefficients of the degree of patent cooperation ($PC_{ij}$), and the distance of cooperation ($GD_{ij}$) in Model 4, were positive and significant ($\beta_1$ = 0.140354, p<0.01, $\beta_2$ = 0.086126, p<0.01). This indicates that the closer the patent cooperation, the tighter the ties between organizations in the network, and the more beneficial for organizations to acquire new technologies and promote technology convergence. This is consistent with the findings of previous studies [21, 60, 61]. Meanwhile, the shorter the cooperation distance, the more favorable the flow of technology, which accelerates the frequency of cooperation and helps organizations accelerate technical uptake. However, the study of collaborative distance is controversial. Zhang and Tang argued [62] that the diversity of collaborative distance promotes innovation performance. And we found that the closer the cooperation distance is, the more it can promote technology convergence. The capability of Chinese non-ferrous metal resource recycling technology to innovate may be insufficient, and international cooperation may not prove to be enough. The non-ferrous metal output is highly polluting, at present, and can only be close to the raw material output to establish industrial parks and deal with the secondary use of waste. This makes long distance cooperation more difficult. Regarding H2, we find that the degree of patent cooperation in Model 5 positively moderates the positive U-shaped relationship between the structural hole constraint index (($SH\ square)_{ij} \times PC_{ij}$) and technology convergence ($\beta$ = 0.061336, p<0.1). Fig 9 shows the moderating effects and the

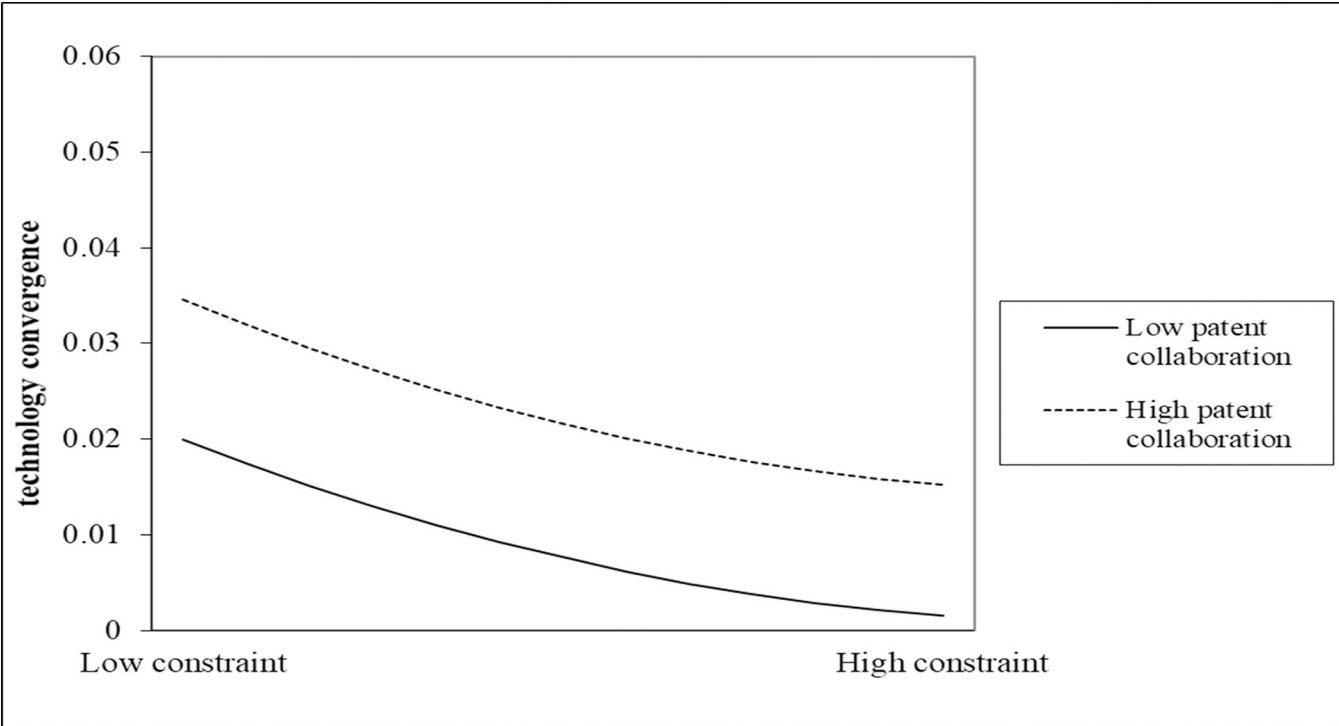

**Fig 9. The moderating effect of degree of patent cooperation network for this study.**

quadratic interaction coefficient between the structural hole constraint index and the degree of patent cooperation. From the figure, the structural hole constraint index shows a positive U-shaped relationship with technology convergence. When the degree of patent cooperation is higher, it strengthens the degree of interdependent ties in the network nodes, which induces the organizational structural hole constraint index to show a positive U-shaped non-linear relationship with technology convergence. Therefore, H2 is verified. This is similar to the findings of previous studies [63]. The greater the degree of cross border cooperation on innovation, the more it helps organizations to occupy a greater number of structural holes and promote innovation. Our study considers the moderating effect of the degree of patent cooperation, on the U-shaped relationship between structural holes and technology convergences. The study shows that the closer the patent cooperation, the higher will be the nodal structural holes in the cooperative network span, accessing more heterogeneous resources and accelerating technology convergence. In addition, the closer the organizations in the network cooperate, the more organizations become relatively less constrained by resources, which reduces the cost of accessing resources and information asymmetry. For example, in the field of non-ferrous metal resource recycling, China established a strategic alliance of technological innovation, which allowed the tenant group to improve its position in the cooperation network and integrate multiple technologies to form a complete industrial chain, including recycling-smelting-reproduction technologies. This result is attributable to the enterprise's long-term industry-university-research cooperation with the Jiangsu Institute of Technology. Fig 9 shows that the bottom of the U-shaped curve shifts upward to the right when the degree of patent cooperation is greater. For example, in the field of non-ferrous metal resource recycling, China established a strategic alliance of technological innovation, which allowed the tenant group to improve its position in the cooperation network and integrate multiple technologies to form a

complete industrial chain, including recycling-smelting-reproduction technologies. This result is attributable to the enterprise's long-term industry-university-research cooperation with the Jiangsu Institute of Technology. Fig 9 shows that the bottom of the U-shaped curve shifts upward to the right, when the degree of patent cooperation is greater.

From Model 6, the cooperation distance negatively regulates the relationship between the structural hole constraint index and technology convergence (β = -0.030539, p<0.1). Fig 10 presents the moderating effect. From the Fig 10, the cooperation distance negatively regulates the relationship between the structural hole constraint index and technology convergence: the closer the cooperation distance, the smaller the structural hole constraint index, the less the node is constrained by external factors, and the more it can promote technology convergence. The farther the cooperation distance, the more the structural holes that must be crossed, and the more the intermediary bridges for the required knowledge technology. Therefore, H3 is verified. This is similar to the findings of a previous study [64]. Guan and Yan showed [26] that the geographical distance between two organizations in a patent collaboration network, has no moderating effect on innovation performance. Instead, we argue that the closer the collaboration distance, the less resource constrained the organization is, and the more quickly it can integrate internal and external technologies and promote technological convergence. It may be related to the fact that non-ferrous metal resource recycling technology field has specificity, non-ferrous metal waste needs to be recycled and treated nearby, and the proximity of cooperation between organizations is preferable, to reduce innovation costs and avoid secondary pollution. The previous research on resource-based theory ignored the effects on the use of resources. This study demonstrates that, the patent cooperation distance is an important context affecting the integration of technology. The reduced distance between the two organizations can prove conducive to mutual technical exchange, and technology convergence. This study further fills the gap of resource-based theoretical research.

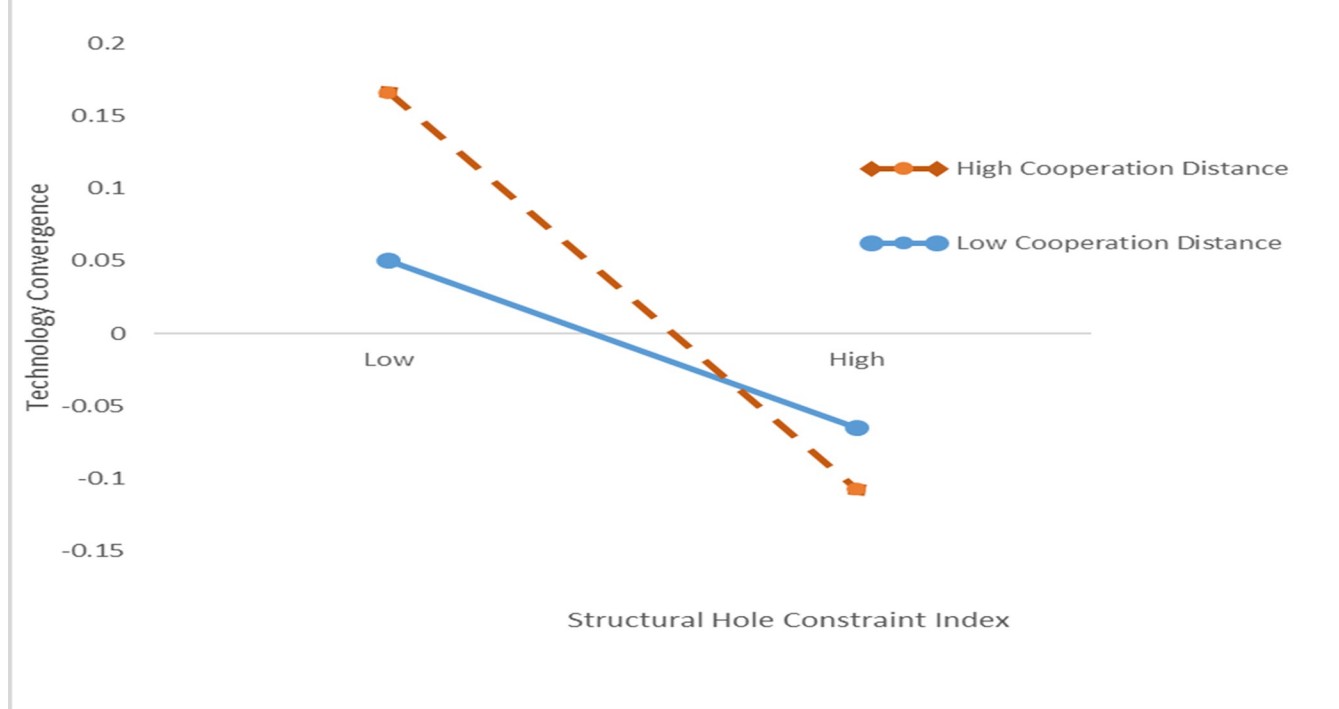

**Fig 10. The moderating effect of distance of patent cooperation network for this study.**

## 5. Discussion

This study employed China's invention patent data of non-ferrous metal resource recycling, to analyze the trend of technological convergence in 52 organizations with the co-occurrence matrices of cooperative patent applications. It examined the non-linear relationship between the structural hole constraint index and technological convergence, in the patent cooperation network, and employed social network theory to verify the moderating role of the degree and distance of patent cooperation. It has been previously noted in the literature [12, 21, 22] that R&D collaboration networks based on patented data, can facilitate technological convergence. However, less literature has considered the contextual factors influencing the impact of structural holes on technological convergence, which are important in influencing internal and external technological convergence, in conjunction with the collaborative network of green technology organizations in China. We fill this gap, and the main research findings are as follows.

This study makes several theoretical contributions to the previous literature. First, we contribute to the technology convergence literature by examining its dynamics for non-ferrous metal resource recycling in China. Previous literature has pointed out that technology convergence, as an important indicator of innovation performance, is also a powerful weapon for the outcome of internal and external technology convergence and for expanding market competition [35]. We found a lack of previous literature examining green technology convergence dynamics. Based on the social network analysis approach, we measure technology convergence mainly based on the IPC co-occurrence matrix. This study collects the non-ferrous metal resource recycling registrations in China based on the 52 types of primary IPC, involved in the non-ferrous metal resource recycling, where the value of two IPCs is the number of associated patents. Previous studies have only focused on the measurement of technology convergence in 3D printing [20], textile [35], and ICT [55] fields. In contrast, this study analyzed the Chinese non-ferrous metal resource recycling technology convergence, using normalized degree, closeness, and betweenness social network analysis indicators based on the construction of IPC co-occurrence matrix. This study found that the non-ferrous metal resource recycling in the core of the network are B03D101/02, B03D101/06, and B03D103/02. Given the intermediary position of C22B7/00 and B03B7/00, relative to other technology areas, the main cooperation advantage of China's non-ferrous metal resources recycling technology is concentrated in the recycling of scrap metal. Equipment and technology methods for processing, scrap metal refining, and conversion and utilization are located at the fringes of the network. Thus, technology convergence proving insufficient, and organizations must improve the refining technology of scrap metal for effective moderation to other technology areas. This study may lead scholars to focus on the dynamics of technology convergence networks within the field of green technology convergence.

Second, this study proposes a new framework for exploring the drivers of technology convergence. We propose that the degree of structural hole constraints formed by inter-organizational patent cooperation affects technology convergence. We use QAP analysis to explore the relationship between structural hole and technology convergence, based on the construction of the structural hole constraint index matrix, and technology convergence matrix. We found that the structural hole constraint index shows a positive U-shaped effect on technology convergence. Organizations in cooperative networks occupy rich structural holes at the beginning of cooperation. They increasingly serve as intermediary bridges, conducive to technology cooperation and knowledge flow transfer. Our study extends the previous literature, and is consistent with the view that richer structural holes improve firm innovation performance [61]. However, previous literature has not examined the linear relationship between structural

holes and technology convergence, and we argue that a moderate but not excessive number of structural holes in collaborative networks, would help organizations to better integrate technology. As cooperation deepens, organizations must cross more structural holes to obtain non-redundant resources and face increased risks and costs. Further, organizations increasingly rely on cooperative relationships, establish trusting cooperative relationships, conduct resource sharing, reduce the cost of searching for information, and effectively filter redundant information. As technological development increases in complexity and market requirements for technology efficacy increase, China's non-ferrous resource recycling technology requires integrating multiple innovations such as recycling processing technology, recycling equipment, and waste refining. Long-term partnerships should be established between organizations to promote technological convergence. Meanwhile, the organization should shift its focus from the number of organizations involved in cooperation to the organizational structure in the industry chain. They should aim to allocate different R&D subjects in the upper, middle, and lower reaches of the technology chain to avoid generating excessive homogeneous technologies. Organizations should implement the industry-academia-research cooperation model and first consider the same type of organizational cooperation, conducive to the rapid transfer of knowledge and technology, given similar organizational structure, integration, and embedding in a larger cooperative network in small groups.

Third, the structural holes based on patent cooperation networks alone do not capture the full picture. The effect of structural holes on technology convergence may depend on two other contextual factors, namely the degree of patent cooperation and the distance of cooperation. We found that the degree of patent cooperation has a significant positive effect on the positive U-shaped relationship between the structural hole constraint index and technology convergence. Our findings enrich previous studies that innovation performance is jointly influenced by the contextual factors of structural capital and the degree of cooperation [63]. Thus, the smaller the constraint index in the patent cooperation network, the richer the structural capital, the cooperation can induce technology convergence. In addition, the cooperation distance should be emphasized in the innovation process and R&D cooperation. We found that the cooperation distance negatively regulates among the structural hole constraint index and technology convergence. The closer the cooperation distance effective the technology export to other members to acquire more resources, and induce more organizations to cooperate willingly. This is controversial with previous views [26, 62]. As mentioned earlier, Guan and Yan's [26] study has not been able to show that geographical distance of organizational cooperation plays a significant moderating role in organizational technological similarity affecting recombine innovation performance. Boschma and Frenken also argue [65] that geographical distance is not a necessary and sufficient requirement for innovation and knowledge sharing today. This differs from the conclusion reached in our study, where we speculate that non-ferrous metal resource recycling technological innovation is different from other industries such as new energy vehicles, artificial intelligence, and ICT, where the operational costs are higher and technological innovation is more expensive if the post use waste of non-ferrous metals is moved and transported to more distant plants and organizations for disposal. It is important for organizations to dispose of the related waste close to the site to save the cost and improve the efficiency of resource use, avoiding secondary pollution and huge waste of resources. In addition, the organizational cooperation system for related technologies in China is not mature, and the promotion of innovations and international cooperation needs to be improved. At present, it is basically a small group cooperating with each other in close proximity, which is conducive to the dissemination of knowledge and technology exchange, and can also apply and absorb new technologies to improve the refining technology after the recovery of non-ferrous metal waste, thus increasing the recyclable use of resources. Conversely, the

geographical distance may bring synergistic effects and difficulties in knowledge transfer [66]. Therefore, it is necessary to absorb more organizations in the cooperative network to participate in cooperative innovation, strike a balance between independent innovation and cooperative innovation, and help organizations conduct technology convergence.

## 6. Conclusion and insights

First, few studies focus on renewable technology, vital for sustainable development, environmental protection, and recycling. The total utilisation of waste resources in China reaches 1.9 billion tons, of which one fifth to one third are used for non-ferrous metals. The recycling technology industry regarding non-ferrous metal resources is indispensable for China's sustainable development. Therefore, this study is significant in that regard. Many studies find that technology convergence is an important means of innovation, but few studies probe technology convergence at the organizational level. Although some examine technology convergence in enterprises [22], the sample was not diverse. Contrarily, this study examines universities, research institutions, and enterprises to study technology convergence precisely. Moreover, the IPC classification number was precise to seven digits, and the Jaccard index was used to measure the affinity, degree of association, and similarity of technologies among individual IPCs. The clustering analysis of co-occurring IPCs with CONCOR was divided into eight segments, and the integration of each segment was measured to analyze the core segments and fringes and depict the structural characteristics and the degree of integration of inter-organizational technology convergence to provide ideas for organizational technology development strategies.

Second, even though some studies show that inter-enterprise cooperation positively affects technology convergence, dividing the types of cooperation into enterprise-university, enterprise-enterprise, and enterprise-university-research institutions [67], they fail to break down the structural characteristics of the cooperation network. They employ linear regression analysis, which does not consider the influence of cooperation network characteristics on technology convergence. Thus, this study specifically subdivides the patent cooperation network into the degree of patent cooperation and cooperation distance and studies the influencing factors on technology convergence. It highlights a path for organizations to seek partners for innovation, suggests how inter-organizational patent cooperation can more effectively improve technology convergence, and enriches and expands the technology convergence literature.

Third, existing studies on technology convergence have not been linked to the structural capital formed by inter-organizational collaborative relationships regarding patents [11]. Data they employ to measure structural capital are based on many subjective questionnaire studies. However, this study employs the constraint index in the structural hole in social networks as the structural hole measurement index, constructs a structural hole constraint based on patent cooperation network and technology convergence theoretical framework, employs the inter-organizational patent cooperation data to calculate the constraint index in the Chinese context, and embeds the degree of patent cooperation, with cooperation distance as moderating variables in the study of technology convergence. The degree of patent cooperation and cooperation distance are embedded as moderating variables in the technology convergence study. The structural hole constraint index shows a non-linear positive U-shaped relationship with technology convergence. Research shows that patent cooperation has a positive or negative effect on firm performance [68], structural holes have a positive or negative effect on innovation output [39, 41], and patent cooperation can change the environment in which firms are located for competitive resources [48]. However, this study finds that patent cooperation has a moderating effect; the richer the subject occupying structural holes in the patent cooperation

network, the more it can dominate the cooperation and obtain more heterogeneous resources, and, if there are more such subjects, the stronger the cohesion of cooperation, which strengthens the cooperation ability and effectively enhances the technology convergence ability.

Finally, this study provides insights to managers and technological policymakers. For managers, the results of the study provide theoretical support, for strategizing about innovation and partner selection of organizations in R&D collaboration networks, thereby promoting technological innovation capabilities. When collaborating on patents, this study shows that universities, leading the cooperation with more structural capital, have more patents in non-ferrous metal resource recycling. Thus, managers should encourage industry-academia-research or patent alliances. Further, same- (different-) organization cooperation is more (less) conducive to technological convergence capabilities, which is consistent with prior findings about a Korean ICT firm cooperation being more conducive to technological convergence than firm-university cooperation [67]. When seeking partners, managers should prioritize cooperation between geographically close organizations and inter-organizational environment, to expand its scope and access to resources. Through initial small cooperation, they should absorb organizations with rich structural holes into the network, balance independent and cooperative innovations, and form a competitive advantage in a larger network. For technological policymakers, they should designate policies to promote cooperation, with universities in the leading role to form a complete cooperation chain, cultivate core innovation subjects, and increase structural capital. The technological policymakers should encourage organizations to build innovation ecosystems. In lieu of non-ferrous metal resource recycling in China, government agencies should support monopolies, and enterprises with strong technological innovation capabilities, as organizations that span more structural holes in R&D cooperation networks, and promote good cooperation with other heterogeneous organizations (especially universities). The technological policymakers need to guide organizations toward cross-border collaboration, to develop key technologies for resource recycling, and to provide the necessary funding or preferential policies. In addition, the government needs to encourage organizations to collaborate with firms in unrelated technological fields, to explore new technologies and conduct effective technological convergence.

Despite the meaningful research contributions, there are some limitations. First, the database platform of China's key industries for patent data analysis comprises only patent data, which is not adequately comprehensive. Future research can incorporate data from news, reports, and history. Second, data on inter-organizational cooperation employs only patent cooperation, which cannot reflect all the situations of inter-organizational cooperation; hence, more cooperation indicators should be considered in future studies. Third, the study considers the impact of the patent cooperation network on technology convergence without considering indicators, such as the density of the network, and some of the proposed findings may not be applicable to other technology industries. Therefore, the research design should be applied to the study of other technology fields.

## Author Contributions

**Conceptualization:** Kai Luo.

**Data curation:** Kai Luo.

**Funding acquisition:** Kai Luo.

**Methodology:** Kai Luo.

**Software:** Kai Luo.

**Supervision:** Kai Luo.

**Writing – original draft:** Kai Luo.

**Writing – review & editing:** Shutter Zor.

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
