## [Decision Letter · Decision Letter 0]

11 May 2022

PONE-D-22-11322China's Non-Ferrous Metal Resource Recycling Technology Convergence and driving factors: A quadratic assignment procedure analysis based on Patent Collaboration-Based Network Structural HolePLOS ONE

Dear Dr. Zor,

Thank you for submitting your manuscript to PLOS ONE. After careful consideration, we feel that it has merit but does not fully meet PLOS ONE’s publication criteria as it currently stands. Therefore, we invite you to submit a revised version of the manuscript that addresses the points raised during the review process.

Please consider all comments

We look forward to receiving your revised manuscript.

Kind regards,

Ahmed Mancy Mosa, Ph.D.

Academic Editor

PLOS ONE

Journal Requirements:

3. Please ensure that you include a title page within your main document. You should list all authors and all affiliations as per our author instructions and clearly indicate the corresponding author.

Reviewers' comments:

Reviewer's Responses to Questions

**Comments to the Author**

1. Is the manuscript technically sound, and do the data support the conclusions?

Reviewer #1: Yes

Reviewer #2: Partly

2. Has the statistical analysis been performed appropriately and rigorously? 

Reviewer #1: Yes

Reviewer #2: Yes

3. Have the authors made all data underlying the findings in their manuscript fully available?

Reviewer #1: Yes

Reviewer #2: No

4. Is the manuscript presented in an intelligible fashion and written in standard English?

Reviewer #1: Yes

Reviewer #2: Yes

5. Review Comments to the Author

Reviewer #1: This paper presents the impact of structural holes in inter-organizational technology cooperation

networks on technology convergence from a patent perspective in China’s non–ferrous metal

resource recycling technology as a case study. Although the concept is nice but it requires a few

modifications to further improve the quality.

1- The introduction part is well researched but the author/s should clearly emphasize the

advantages of the methods used in this study, Introducing the methods alone is not enough and

the necessity of choosing them is also important.

2- In data-driven research, the methodology of data analysis and information about the data used

in the research are important. In the introduction section, The author/s are more concerned about

the application, rather than the data and methodologies.

3- Authors should provide a general figure of the framework of the research at the end of the

introduction section in full detail.

4- The Results (section 4.) and Research method (sub-section 3.3) text part are very weak. It is

true that the methodology is not innovative and is known, but a more accurate introduction of the

methods is needed. The reader does not have any clear information about how the methods work.

Clear articulation of methods and results in light of the Research question or hypothesis is

needed.

5- Results: I suggest you discuss a little bit more the results of the case study with the previous

(and recent) literature. I think you have space here to discuss more your results and findings.

Reviewer #2: 1-The introduction section has been revised and shortened.

2-The author should show the novelty of this research, the result, and the finding this research need requires discussion using relevant theories and previous research.

3-Please show what is the contribution of the research to the parties who need these results in accordance with the objectives of this research.

4-Lastly, the discussion section is somewhat weak. The authors should clearly state the key lessons learned. There is a need to strengthen the argument of the paper, most of your assertions are loosely accompanied by excessive juxtaposing. Please address this issue.

6. PLOS authors have the option to publish the peer review history of their article (what does this mean?). If published, this will include your full peer review and any attached files.

Reviewer #1: No

Reviewer #2: **Yes: **Ahmed A. Zaid

---

## [Author Response · Author response to Decision Letter 0]

24 Jun 2022

Response letter

Dear Editors and Reviewers,

We are extremely grateful for the opportunity to further revise our manuscript, entitled "China's non-ferrous metal recycling technology convergence and driving factors: a quadratic assignment procedure analysis based on patent

collaboration-based network structural hole" (No. PONE-D-22-11322). Our data can be found at https://doi.org/10.7910/DVN/EO2R2R

We are particularly thankful to the referees for their comments, which have guided us in strengthening the manuscript. With the best effort to thoroughly revise this manuscript according to their suggestion, we have preserved the paper’s originality as much as possible.

The revised version and point-by-point responses to the reviewers’ comments are included below. Major changes in this manuscript have been highlighted in red. Thank you for your suggestions, and we hope our revisions will strengthen the paper and that the paper will meet the publication requirements of PLOS ONE.

Kind regards, 

Authors of Manuscript (No. PONE-D-22-11322) 

Revision Outline

Dear editors and reviewers, thank you for the opportunity to revise this article. We have made substantial attempts to refine the existing ideas and mitigate the deficiencies. We will present our modification method in the following sections. First, an outline of the overall modifications is presented and explained their broad scope. Second, the subsequent parts contain answers to the questions raised by the reviewers one by one.

All changes are marked in red in the manuscript. We have adjusted the content layout of the article considering the reviewers' comments, as evidenced by the deletion of redundant sections and the addition of incomplete discussions. We also made changes to the figures in the article to meet the reviewers' suggestions regarding the data process. Later, we formatted the table to ensure it conforms with the table style used in PLOS ONE published papers. The responses to the reviewer's comments are as follows:

Reviewer #1

This paper presents the impact of structural holes in inter-organizational technology cooperation networks on technology convergence from a patent perspective in China’s non–ferrous metal resource recycling technology as a case study. Although the concept is nice but it requires a few modifications to further improve the quality. 

Response: Thank you for your detailed comments, which have further strengthened the manuscript. The paper has been carefully revised, according to your suggestions. Overall, we do hope our responses can meet your expectations.

Comment 1: The introduction part is well researched but the authors should clearly emphasize the advantages of the methods used in this study, Introducing the methods alone is not enough and the necessity of choosing them is also important.

Response: Thank you for your suggestion. Based on your observation, admittedly, we seem to have devoted an excessive amount of space to the methods without explaining the necessity. To rectify the same, we have incorporated the 3rd and 4th paragraphs of the Introduction to address the rationality and the necessity of social networks and QAP methods. Here we explain and address the need for the methods as followings: 

The necessity of using social networks. First, we add literature analysis to explain why the IPC co-occurrence matrix was used. Then we construct a network of technological convergence to explore the dynamics of technology convergence in Non-Ferrous Metal Recycling. We have included a summary of the differences and shortcomings between the previous methods of analyzing technology convergence dynamics and our construction of the IPC co-occurrence matrix. This showcases the necessity and advantages of using this matrix to explore the technology convergence network. Jeong S and Lee S (2015) proposed a method to verify whether technology convergence is based on research and development (R&D) project collaborations. The integrated technologies are relatively similar if the same type of organization is collaborating. Otherwise, if the different types of organizations collaborate, two different kinds of technologies are possibly integrated. We highlighted our approach's rationality and necessity based on the literature analysis. The shortcomings of the aforementioned measurement techniques include difficulties in obtaining the limited data, a lag in the data, and the lack of attention to the integration of new technologies. Therefore, our study chose the co-occurrence of patented data of International Patent Classification (IPC) numbers. In addition, when it comes to the analysis method of patent data, the literature [19-21] uses the Herfindahl index, entropy, and the total number of convergences for technology convergence as measurement. However, we argue that these methods above cannot observe the process and degree of convergence of individual technology nodes in the overall network.

Meanwhile, the absolute number of patents cannot describe the technology convergence dynamics and the different technology areas involved in convergence. A patent document that contains two or more IPC classification numbers implies that the patent involves multiple technologies, reflecting the sources and trends of technologies and applications [22]. Considering the above analysis, we chose social networks to analysis our patent data.

The necessity of using QAP regression. In the 4th paragraph of the Introduction, we justify the need and advantages of using QAP regression to analysis the impact of structural holes based on patent collaboration on technology convergence and related contextual factors of the study. Existing literature, for example, literature [12] discusses R&D collaboration based on the number of R&D project organizations, the output of knowledge, technological diversity, and the numbers of employees are involved in R&D collaboration projects, which fails to reflect the flow of technology and the degree of collaboration and the specific organizational structure of collaboration. Faust and Wasserman (1992) define a "one-mode" network concept [23] and argue that constructing an R&D cooperation matrix based on social networks can overcome the shortcomings of the above studies. The co-present value of two nodes in the network reflects the organizational mobility, the degree of cooperation, and the structural characteristics of the cooperative network. And based on literature [24], we argue the science and necessity of measuring R&D cooperation based on patent cooperation network. Patents are the output of R&D, reflecting the information of technology, the application of technology, and R&D cooperation performance and innovation path. Therefore, we construct an organizational R&D cooperation network based on patent data to study technology convergence factors. The rationality and scientific validity of the method is fully discussed in this section. In addition, to justify the QAP analysis method, we argue that one-model social networks cannot analyze the relationship between individual networks. For example, it is impossible to portray whether each nodal organization in a patent cooperation network plays the role of an intermediary or core collaborator and structural hole (bridge) and whether it has any effect on innovation. At the same time, big data and the development of the network integration process have complicated the network formed by the relationship between individuals and groups. Lastly, the general multiple regression analysis methods cannot explore the non-independent relationships between networks and networks. And the covariance problem makes the least squares (OLS) method invalid. Therefore, we introduce the quadratic assignment procedure (QAP) to verify the impact of structural holes on technology convergence networks under relationships of organized patent cooperation. Meanwhile, QAP also can solve the auto-correlation problem and produce relatively unbiased statistical results [25-27].

Please see the original text below (in red) for the specific changes directly made to the manuscript. 

"Introduction

Therefore, this paper begins by describing the trends in the convergence of the analytical techniques for co-occurrence-based technology in non-ferrous metal recycling. Technology convergence can be divided into the following categories. The first category is based on whether the types of organizations cooperating in R&D projects are identical. If they belong to the same category, the technologies used may be similar, instead of two completely different technologies being integrated together. If the types of cooperating organizations belong to different categories, it is possible that two different types of technologies are integrated together [12]. In addition, some scholars have used a measurement approach to study technology convergence, based on academic literature and input-output (I/O) tables and research data [13-15]. The shortcomings of the aforementioned measurement techniques include difficulties in obtaining the limited data, a lag in the data, the requirement of a long period of observation, and the reflection of the integration of only industries and applications, without reflecting the integration of new technologies. However, the current popular measurement of technology convergence method is based on the co-occurrence of the International Patent Classification (IPC) number, for patented data, fused with social network analysis technology and patent citations [16-18]. In this study, the co-occurrence of patent data and IPC helps in constructing a patent cooperation network, to describe technology convergence of non-ferrous metal recycling based on social network theory. The value of co-occurrence of two IPC classification numbers, in a patent network, represents the number of patents in which these two IPCs occur simultaneously. Previous studies measured the Herfindahl index [19], entropy [20], and total number convergence patents [21] based on patent data, yet could not observe the process and degree of convergence of individual technology nodes in the overall network. The absolute number of patents also could not describe the dynamics of technology convergence, and different technological areas involved in convergence. However, a patent document with two or more IPC classification numbers implies that the patent involves multiple technologies, reflecting the source and development trend of the technologies and their applications [22]. This research constructs a patent network of IPC co-occurrence, based on the data from the patent information platform of key industries, of the State Intellectual Property Office of China. This can observe the composition of technologies, and the degree of integration of different technological nodes in the network. The patented information platform provides the information of non-ferrous metal recycling patents, primarily applied by innovative Chinese organizations. Describing the technological convergence dynamics proves beneficial for this study.

Hence, the second objective of this paper is to employ the social network theory and construct an inter-organizational R&D cooperation network, specifically divided into structural hole constraint index matrices. This analyses the effect of structural holes on technology convergence, under the contextual factors of patent cooperation degree and distance. Previous studies of R&D cooperation were based on the number of organized projects, knowledge output, technological diversity, and the number of employees involved in such R&D projects [12]. These are absolute data, not reflecting the flow of technology, the degree, and the specific organizational structure of cooperation. Faust and Wasserman [23] defines "one-mode" network, which speaks of some scholars measuring R&D cooperation with patent cooperation network. Patents are the R&D outputs that reflect the information and innovation outcomes of technology and its application, performances of collaborated R&D, and innovation paths [24]. Using the social network approach to construct a one-model patent cooperation network, this study observes the number of cooperated patents between two organizations in the network, reflecting the degree of R&D cooperation between them. The mobility of knowledge depends on the flexibility of the overall network, the nature of each nodal organization and the distance between them. However, this one-model social network cannot analyze the relationship between each network. For example, it cannot portray whether each nodal organization in the patent cooperation network plays the role of an intermediary or a core collaborator. It also cannot portray whether a structural hole (bridge) in the network, has any effect on the performance of innovation. At the same time, the advent of the era of big data and the development of the network integration processes have complicated the network formed by the relationship between individuals and groups. The general multiple regression analysis method cannot explore the non-independent relationships between different networks, and the covariance problem makes the ordinary least squares (OLS) method, based on time series and panel data, invalid. In this study, we try to use the quadratic assignment procedure (QAP) test to examine the hypothesis of the "relationship-relationship,"and analyze the relationship data of the organization's patent cooperation. This method is based on the QAP test, which helps examine the relationship data under the organizational patent partnership and other factors affecting technological convergence, solving the auto-correlation problem, and producing relatively unbiased statistical results [25-27]."

Comment 2: In data-driven research, the methodology of data analysis and information about the data used in the research are important. In the introduction section, the authors are more concerned about the application, rather than the data and methodologies.

Response: Thank you for the comments. In continuation with your suggestion, in the third paragraph of the Introduction, we specifically state that our study is based on patent data to analysis the impact of structural holes on technology convergence. Since a structural hole is based on the organization’s patent cooperation network, the technology convergence network, the degree of patent cooperation, and the cooperation distance all involve patents. Therefore, the constructed co-occurrence value of two such organizations is the number of cooperating patent applications. The IPC in the technological convergence network represents the technology area involved in non-ferrous metal recycling. The co-occurrence value of two IPCs is the number of co-associated patents. The larger the number of patents reflects the degree of technological convergence. Our patent data are from the patent information platform of key industries of the State Intellectual Property Office of China. Based on the above data, we construct IPC co-occurrence. In the end, we add the sentences that introduce the QAP regression analysis method applied to the analytical process of our study.

Second, in the 4th paragraph of the Introduction section, we introduced the QAP regression analysis method. The social network approach is used to construct a one-model patent cooperation network. The number of cooperation patents between two organizations in the network reflects the degree of R&D cooperation between organizations. The mobility of knowledge can also indicate the degree to which the overall network is flexible, the nature of each node organization, and the distance between organizations. However, this one-model social network cannot analyze the relationship between each network. For example, it cannot portray whether each node organization in the patent cooperation network plays the role of an intermediary or a core collaborator and observes whether the network's structural hole (bridge) affects innovation performance. At the same time, big data and the development of the network integration process have complicated the network formed by the relationship between individuals and groups. The general multiple regression analysis methods cannot explore the non-independent relationships between networks and networks. The covariance problem makes the ordinary least squares (OLS) method invalid. In this paper, we use the quadratic assignment procedure (QAP) to examine the hypothesis.

Please see the manuscript below (in red) for the specific changes. 

"However, prior studies show lesser engagement with China’s non-ferrous metal resource recycling technology. Furthermore, few studies explore the mechanism of structural holes’ influence on technology convergence. Therefore, this paper begins by describing the trends in the convergence of the analytical techniques for co-occurrence-based technology in non-ferrous metal recycling. Technology convergence can be divided into the following categories. The first category is based on whether the types of organizations cooperating in R&D projects are identical. If they belong to the same category, the technologies used may be similar, instead of two completely different technologies being integrated together. If the types of cooperating organizations belong to different categories, it is possible that two different types of technologies are integrated together [12]. In addition, some scholars have used a measurement approach to study technology convergence, based on academic literature and input-output (I/O) tables and research data [13-15]. The shortcomings of the aforementioned measurement techniques include difficulties in obtaining the limited data, a lag in the data, the requirement of a long period of observation, and the reflection of the integration of only industries and applications, without reflecting the integration of new technologies. However, the current popular measurement of technology convergence method is based on the co-occurrence of the International Patent Classification (IPC) number, for patented data, fused with social network analysis technology and patent citations [16-18]. In this study, the co-occurrence of patent data and IPC helps in constructing a patent cooperation network, to describe technology convergence of non-ferrous metal recycling based on social network theory. The value of co-occurrence of two IPC classification numbers, in a patent network, represents the number of patents in which these two IPCs occur simultaneously. Previous studies measured the Herfindahl index [19], entropy [20], and total number convergence patents [21] based on patent data, yet could not observe the process and degree of convergence of individual technology nodes in the overall network. The absolute number of patents also could not describe the dynamics of technology convergence, and different technological areas involved in convergence. However, a patent document with two or more IPC classification numbers implies that the patent involves multiple technologies, reflecting the source and development trend of the technologies and their applications [22]. This research constructs a patent network of IPC co-occurrence, based on the data from the patent information platform of key industries, of the State Intellectual Property Office of China. This can observe the composition of technologies, and the degree of integration of different technological nodes in the network. The patented information platform provides the information of non-ferrous metal recycling patents, primarily applied by innovative Chinese organizations. Describing the technological convergence dynamics proves beneficial for this study.

Hence, the second objective of this paper is to employ the social network theory and construct an inter-organizational R&D cooperation network, specifically divided into structural hole constraint index matrices. This analyses the effect of structural holes on technology convergence, under the contextual factors of patent cooperation degree and distance. Previous studies of R&D cooperation were based on the number of organized projects, knowledge output, technological diversity, and the number of employees involved in such R&D projects [12]. These are absolute data, not reflecting the flow of technology, the degree, and the specific organizational structure of cooperation. Faust and Wasserman [23] defines "one-mode" network, which speaks of some scholars measuring R&D cooperation with patent cooperation network. Patents are the R&D outputs that reflect the information and innovation outcomes of technology and its application, performances of collaborated R&D, and innovation paths [24]. Using the social network approach to construct a one-model patent cooperation network, this study observes the number of cooperated patents between two organizations in the network, reflecting the degree of R&D cooperation between them. The mobility of knowledge depends on the flexibility of the overall network, the nature of each nodal organization and the distance between them. However, this one-model social network cannot analyze the relationship between each network. For example, it cannot portray whether each nodal organization in the patent cooperation network plays the role of an intermediary or a core collaborator. It also cannot portray whether a structural hole (bridge) in the network, has any effect on the performance of innovation. At the same time, the advent of the era of big data and the development of the network integration processes have complicated the network formed by the relationship between individuals and groups. The general multiple regression analysis method cannot explore the non-independent relationships between different networks, and the covariance problem makes the ordinary least squares (OLS) method, based on time series and panel data, invalid. In this study, we try to use the quadratic assignment procedure (QAP) test to examine the hypothesis of the "relationship-relationship," and analyze the relationship data of the organization's patent cooperation. This method is based on the QAP test, which helps examine the relationship data under the organizational patent partnership and other factors affecting technological convergence, solving the auto-correlation problem, and producing relatively unbiased statistical results [25-27]."

Comment 3: Authors should provide a general figure of the framework of the research at the end of the introduction section in full detail.

Response: Thank you for your comments. Combined with your further suggestion, we have constructed a figure of the framework and added it to the end of the introduction section.

 Fig 1. The framework of Research

Comment 4: The Results (section 4) and Research method (sub-section 3.3) text part are very weak. It is true that the methodology is not innovative and is known, but a more accurate introduction of the methods is needed. The reader does not have any clear information about how the methods work. Clear articulation of methods and results in light of the Research question or hypothesis is needed.

Response: Thank you for reminding us about this important point. Combined with this suggestion (Comment 4), in the Research methods and model, we have refined the information on the working principle of the methodology, with its clear articulation and definition. The original Results (section 4) section is revised as the Empirical results and analysis section. In the first paragraph, to analysis the degree of integration of non-ferrous metal resource recycling technologies, we have selected the primary 52 IPCs involved in these technologies. To obtain patent applications containing these 52 IPC classification numbers, we searched the required patents in the Patent Information Service Platform for Key Industries of the State Intellectual Property Office of China, spanning the period 1985-2019. Based on the patented technologies associated with these 52 IPC classification numbers, in the second paragraph of this section, we construct the IPC co-occurrence matrix Jij, and explain the detailed meaning of the two nodes and their values in this matrix. Then, we calculate Jij based on , to construct the overall technology convergence matrix. Subsequently, we constructed three indicators based on the social network analysis method, Normalized Degree, Closeness, and Betweenness, to analysis the convergence trend. We visualized the technology convergence network by using Gephi software. We also added literature [25] to illustrate the meaning and scientific application of the normalized degree and between indicators. Literature [56,57] show the closeness value reflects the distance of nodes in the fusion network to disseminate information. The larger the value, the shorter the distance of the corresponding node from other nodes, which indicates better mobility and faster fusion of the technology. In addition, further analysis of the technology fusion, we illustrate in the main text the hierarchical clustering analysis of the network nodes using CONCOR, a hierarchical clustering analysis method in social network analysis. Hierarchical clustering analysis of nodes in a network can observe local convergence trends in technology areas in the network and reveal the degree of convergence and the relative position and role of each subdivision of technology in the network partitioned into modules after CONCOR clustering.

To explore the effect of structural holes based on inter-organizational patent cooperation network on Jij matrix influence, we illustrate in the third paragraph of this section that this paper adopts QAP regression with Jij as the dependent variable. The main 52 cooperative patent application organizations are extracted from the literature mentioned above, collected from the database of the patent information service platform of the State Intellectual Property Office of China. The social network analysis method is used to construct the patent cooperation matrix PCij, and the meaning of the double node values is illustrated in the construction of PCij based on , to calculate the structural hole constraint index among organizations, thus constructing the constraint index matrix SHij as the independent variable. In addition, to study the changes of structural hole’s influence on technology convergence under the contextual factors of patent cooperation, this paper takes PCij as a matrix to measure the degree of patent cooperation, and transforms the inter-organizational values of vertical and horizontal axes in the network into cooperation distance values, which are 1 if two organizations are in the same province and 0 otherwise, to construct the distance matrix GDij of patent cooperation. We take PCij and GDij as two contextual factors of patent cooperation, and the above SHij and Jij matrices are subjected to QAP regression analysis and the interaction matrices are constructed by combining SHij and (SH square)ij terms with PCij and GDij, respectively, to analyze the moderating effects of the degree and distance of patent cooperation. At the end of this section, we detail the rationality and scientific validity of the analysis using QAP regression, and justify it based on the literature [25,27,58]. Further introduction of the process and advantages of QAP regression, provides the regression model and makes detailed explanations of the variables in the fourth paragraph. 

Details can be found in the manuscript’s Research methods and model section, and the changes have been marked in red. 

"Research methods and model

In order to analyze the degree of technological convergence among non-ferrous metal recycling technologies, we selected the main 52 IPC classifications involved in these technologies. Patent applications containing these 52 IPC classification numbers were obtained from the Patent Information Service Platform for Key Industries of the State Intellectual Property Office of China, spanning the period of 1985–2019. This platform has a dedicated non-ferrous metal recycling technology patent database, which contains all information about the relevant patented documents filed in China, such as IPC classification, year of filing, number of patents under each sub-IPC classification, major patent applicants and co-innovators, and number of co-filed patents. These patents help us observe the trend of the number of these patents, that of the associated IPC patents, and the main co-innovators filing patents.

The construction of the IPC co-occurrence matrix Jij is based on the patented technologies associated with these 52 IPC classification numbers. In this matrix, the vertical and horizontal axes are the fields of non-ferrous metal recycling technology subdivision, and the two technological field elements are the number of fused patents containing the two common IPC classification numbers. The Jaccard index of the two nodes in the matrix is calculated according to Equation (1), and the overall technological convergence matrix is constructed. In order to analyze the overall network and the convergence trend of each technology area in the network, we analyze the normalized degree, closeness, and betweenness of the nodes in the network based on the social network approach to measure the role of each technology area in the convergence network and visualize the Jij. matrix using Gephi software to make the technology convergence more intuitive. The normalized degree of a node is the number of nodes directly connected to that node. The evolution of the network structure is explained by building a model of technology convergence network formation. The degree factor, as a key factor affecting the evolution of the network, reflects the core and key technologies of fusion [25]. Closeness can determine the distance of nodes in the fusion network to disseminate information [56,57], and the larger the value, the shorter the distance from the node corresponding to that value to other nodes, indicating better mobility and faster fusion of technologies. Betweenness indicates the medium ability to transfer knowledge, which can be described as the potential influence of a technology, and if a technology has high intermediary centrality, it can be defined as a medium with high potential to facilitate knowledge transfer in different domains. In addition, to further analyze this technology convergence, we perform hierarchical cluster analysis of network nodes using CONCOR, a hierarchical cluster analysis method in social network analysis. Hierarchical cluster analysis of the nodes in the network can observe the local convergence trends of the technology domains in the network, and can reveal the degree of convergence and the relative position and role of each subdivision of the technology after CONCOR clustering when the network is partitioned into modules [25].

In order to explore the structural hole based on inter-organizational patent cooperation network on Jij. technology convergence matrix influence effect, we used QAP regression with Jij. as the dependent variable. The main 52 cooperative patent application organizations were extracted from the above-mentioned patent literature collected in the patent database of non-ferrous metal recycling technology of the patent information service platform of the State Intellectual Property Office of China, and the patent cooperation matrix PCij was constructed using the social network analysis method, which is a one mode matrix with the horizontal and vertical axis nodes represent organizations. The larger the value, the closer the degree of cooperation between the organizations, the more resources they obtain from each other, and the closer the overall network. Based on the construction of PCij, the structural hole constraint index between organizations is calculated based on equation (2), as to construct the structural hole constraint index matrix SHij of organizational patent cooperation as the independent variable to study the influence effect of structural hole on technology convergence. In addition, in order to study the changes of structural hole influence on technology convergence under the contextual factors of patent cooperation, this paper takes PCij as a matrix to measure the degree of patent cooperation and transforms the inter-organizational values of vertical and horizontal axes in the network into cooperation distance values, which are 1 if two organizations are in the same province, and 0 otherwise, to construct the distance matrix GDij of patent cooperation. We take PCij and GDij as two contextual factors of patent cooperation, and the above SHij and Jij matrices are subjected to QAP regression analysis and the interaction matrices are constructed by combining SHij and SHij quadratic terms with PCij and GDij, respectively, to analyze the moderating effects of cooperation degree and cooperation distance of patent cooperation. Since the above variables are network binary data, not panel and time series data, it is difficult to use OLS regression and each variable must be independent and positively distributed. However, the nodes in the network data are interrelated with each other and have potential indirect or direct dependencies. Therefore, the assumptions of OLS regression are not satisfied. Therefore, QAP regression is used instead in this paper. This regression method uses non-parametric alignment, and in QAP analysis, the correlation coefficients of the independent and dependent variable matrices are derived after multiple rounds of serial transformations and iterations of the vertical and horizontal axes in the network, and a test statistic is obtained to test the original hypothesis of the regression equation and whether the significance level rejects the original hypothesis. When the degree of auto-correlation between variables is high, the use of QAP has a smaller percentage of errors than the OLS [25]. QAP is a research method for analyzing the relationship between each co-occurrence matrix. The study employs the QAP for regression analysis based on the analysis of technological convergence for several reasons. It solves the problem of auto-correlation between variables, allows for a method of comparing the similarity of each element in two matrices, gives the correlation coefficient between the two matrices, performs a non-parametric test, and generates relatively unbiased statistical results applicable to this study [27,58]. The regression equations in this paper are shown in equation (3).

 (3)

In equation (3), Jij represents the technological convergence matrix, SHij represents the structural hole constraint index matrix, PCij represents the patent cooperation matrix, GDij is the cooperation distance matrix, PSij represents the patent stock matrix, and OSij represents the organizational types matrix. Equation (3) follows the method for QAP regression, with the same meaning represented through the variables. Based on the results of the QAP regression, we can test the first to third hypotheses, on the grounds of coefficients and significance levels of their independent variables. "

Second, in the Empirical results and analysis section, we have improved the discussion of the analysis of the results, and how it proves the hypothesis of our study, in correspondence with the analysis indicators and research methods proposed in the Research Methods and Model section. We clearly explain the principles of the methodology, the empirical analysis and its results in detail. The process through which the QAP regression results prove the research hypothesis, is meticulously discussed in the results. In the first paragraph of this section, we analysis the dynamics of organizational patent cooperation. We use Ucinet software to calculate the Normalized Degree of each organization node in the constructed patent cooperation matrix PCij and the ranking based on the Normalized Degree value. Based on the results of Normalized Degree analysis, we focused on "Jiangxi University of Science and Technology," "Central South University, " and "Western Mining Co., Ltd." and draws relevant conclusions.

In the second paragraph of this section, we use Ucinet software to calculate the values of Normalized Degree, Closeness, and Betweenness for the constructed IPC co-occurrence matrix. We focus on the values of Normalized Degree, B03D101/02 (17.347), B03D101/06 (12.465) and B03D103/02 (10.904), and explain these three IPCs and draw relevant conclusions. We also analyzed the closeness value in the technology convergence network and provided an in-depth analysis of the meaning of this indicator and the technology convergence dynamics reflected by it, and drew relevant conclusions. We further analysis the betweenness values in the technology convergence network and describe in detail the meaning of the values reflected by the results of the calculations, focusing on the three IPCs C22B7/00 (4.092), B03B7/00 (3.804) and C22B7/04 (3.033), and elaborate on the meaning of the betweenness The value of the three IPCs represents the meaning of "working-up raw materials other than ores (e.g., scrap, to produce non-ferrous metals or compounds thereof)," as well as the meaning of "working-up raw materials other than ores (e.g., scrap, to produce non-ferrous metals or compounds thereof)," in China Non-Ferrous Metal Resource Recycling. metals or compounds thereof)," "Combinations of wet processes or apparatus with other processes or apparatus (e.g., for dressing ores or garbage)," and "Working up slag" are discussed in terms of convergence trends and future development of technical areas. In addition, we use Gephi software to visualize the IPC co-occurrence matrix and illustrate the meaning of nodes and lines in the visualization diagram.

In the CONCOR Analysis section, we added a part of the hierarchical analysis, which further specifies that, Module 1 and Module 2 have the highest association density, indicating that in the technology convergence network, the elements in Module 1 are most closely linked with those in Module 2. This reflects a higher degree of convergence between the technological domains involved in Module 1 and Module 2, mainly for B03D and the convergence of the IPC subcategories under the B03B category. China has a strong innovation capability in the technology areas of recycling non-ferrous metal resources for flotation, selective deposition method and separation of solid materials, by liquid, by wind shaker or wind jigger. The country has formed very mature and stable fusion technologies, in these two areas.

Finally, in the QAP Analysis section, we focus on improving the analysis and discussion of the regression results, highlighting the innovative and research ideas of the paper, and clarifying the meaning of the regression coefficients, how the QAP regression verifies the research hypothesis, and the underlying influence. In the first paragraph of the Regression Results section, we analysis the relationship between the structural hole constraint index SHij and Jaccard index matrix Jij constructed by QAP, and then analysis the mechanism of the influence of the structural hole on technology convergence based on patent cooperation network. We clearly explain the principle and process of QAP regression analysis, and estimate seven models, detailing which variables are included in each model. Then, starting in the third paragraph of this section, we analysis how the regression results test the hypotheses and discuss the regression results from models 1-6 in turn. For example, models 2 and 3 from the regression results show that the regression coefficients of the structural hole constraint index matrix (SHij) and the quadratic structural hole constraint index matrix ((SH square)ij) are negative, positive and significant, respectively (β1=-0.068671, p<0.01, β2=0.068968, p<0.05). We conducted a comparative analysis with the literature [59], and found that, the more structural holes a firm observes in an R&D network, the lesser constrained it is by the connected nodes, and the more complementary unrelated diversified knowledge it can acquire. This effectively absorbs and integrates internal and external knowledge, and promotes the firm's exploitative and exploratory innovation outcomes. However, the innovation performance of the above studies is relatively homogeneous, and does not consider the linear relationship between structural holes and innovation performance. Instead, we examine the linear relationship between the structural hole and technological convergence from the technology convergence perspective, and draw a richer conclusion.

In the fourth paragraph of this section, to validate H2, we consider adding the degree of patent cooperation (PCij), and cooperation distance (GDij) to Model 4, with positive and significant regression coefficients (β1=0.140354, p<0.01, β2=0.086126, p<0.01). Adding literature [62] for comparative discussion, enriches the findings and interprets the empirical results with specific cases. Regarding H2, we find that the degree of patent cooperation in Model 5, favorably moderates the positive U-shaped relationship between the structural hole constraint index ((SH square)ij×PCij), and 

technological convergence (β=0.061336, p<0.1). Further validation of H2 is done in conjunction with Figure 9. A discussion section is added in this portion of the manuscript, in conjunction with the literature [63].

In the fifth paragraph of this section, combined with Model 6, the cooperation distance negatively regulates the relationship between the structural hole constraint index, and technology convergence (β=-0.030539, p<0.1). Figure 10 presents the moderating effect. Therefore, H3 is verified. Interestingly, our findings are contrary to the literature [26], which shows that the geographical distance between two organizations in a patent collaboration network, has no moderating effect on innovation performance. Instead, we conclude that the closer the collaborative distance, the less resource-constrained the organization is and the more rapidly it can integrate internal and external technologies and promote their convergence. In this regard, we provide a rational explanation.

The changes we made to the Empirical Results and Analysis section, are highlighted in red in the main manuscript. (Highlighted in red)

"Empirical results and analysis

Patent cooperation network and technology convergence analysis

We used Ucinet software to calculate the normalized degree of each organization node in the constructed patent cooperation matrix PCij and the ranking based on the normalized degree value. Patent applications by universities dominate that of enterprises in technology innovation regarding non-ferrous metal resources recycling in China. Hence, there is room for improvement in innovation and transformation applications. Table 2 highlights the specific classification of each organization. According to the analysis results of the normalized degree, "Jiangxi University of Science and Technology," "Central South University, "and Western Mining Co., Ltd." ranked in the top three, with "Jiangxi University of Science and Technology" having the highest normalized degree value (3.922). This indicates that in the patent cooperation network, "Jiangxi University of Science and Technology" has the most connected nodes, the greatest degree of cooperation with other organizations, and the strongest technological innovation ability, and is in the core position in the network. Otherwise, from Table 2, there are 10 universities, 13 research institutions, and 29 enterprises; although universities were least represented in the sample, they have a larger normalized degree and are crucial in the cooperation network. Universities pay more attention to open innovation and effective integration of technology. Further, visualizing the patent cooperation matrix PCij with Gephi software, as shown in Fig 6, the nodes represent each organization and the co-occurring patents; the thickness of the connecting lines connecting the nodes indicate the number of co-occurring patents. In Fig 6, the size of each node indicates the degree of cooperation between the organizations, and the thickness of each path indicates the number of patents of cooperation between organizations; the higher the frequency of cooperation, the thicker the line of the path. Research institutions cooperate in greater numbers than enterprises, although their degree is smaller. Further, institutions are more capable of independent innovation (Fig 6).

Table 2. Classification of major patent cooperative organizations.

Types of Organizations Name Normalized Degree Rank

Colleges and Universities (10) Jiangxi University of Science and Technology 3.922 1

 Central South University 0.218 2

 Kunming University of Science and Technology 0.436 7

 Hebei University of Engineering 2.397 9

 Wuhan Institute of Technology 1.089 14

 Guizhou Institute of Technology 1.743 17

 Central South University 1.089 29

 Guangxi University 0.218 36

 Taiyuan University of Technology 0.871 37

 University of Science and Technology Liaoning 0.218 38

Research Institutes (13) Hunan Institute of Non-Ferrous Metals 1.307 4

 Institute of Process Engineering, Chinese Academy of Sciences 1.089 5

 Guangdong Institute of Comprehensive Utilization of Resources 0.218 13

 China Coal Geological Engineering Co., Ltd. Beijing Institute of Hydraulic Engineering and Environmental Geology 1.743 19

 Beijing General Research Institute of Mining and Metallurgy 0.218 25

 Guangzhou Institute of Geochemistry, Chinese Academy of Sciences 1.307 27

 Changchun Gold Research Institute 1.089 31

 China Gold Group Corporation Technology Center 1.089 32

 Zhengzhou Zhongke Emerging Industry Technology Research Institute 1.089 33

 Shenyang Aluminium Magnesium Design & Research Institute 0.871 39

 Jilin Provincial Metallurgical Research Institute 0.654 50

 Zhengzhou Light Metal Research Institute 0.871 51

 China Non-ferrous Metals Technology Development Exchange Center 0.871 52

Enterprises (29) Western Mining Co., Ltd. 2.832 3

 Western Mining Group Technology Development Co., Ltd. 2.397 6

 Zijin Mining Group Co., Ltd. 1.961 8

 Daye Non-Ferrous Design and Research Institute Co., Ltd. 1.961 10

 Shenzhen Zhongjin Lingnan Non-ferrous Metals Co., Ltd. 1.307 11

 Tianjin Shunneng Shijia Environmental Protection Technology Co., Ltd. 2.179 12

 Xiamen Zijin Mining and Smelting Technology Co., Ltd. 1.961 15

 Jilin Haorong Technology Development Co., Ltd. 1.743 16

 Daye Non-ferrous Metals Co., Ltd. 1.743 18

 Great Wall Aluminum Company of China 1.089 20

 Guizhou Guifu Ecological Fertilizer Co., Ltd. 1.743 21

 Guizhou Kexin Chemical and Metallurgical Co., Ltd. 1.743 22

 Jilin Gene Nickel Co., Ltd. 1.525 23

 Beijing Building Materials Science Research Institute Co., Ltd. 0.654 24

 China Ruilin Engineering Technology Co., Ltd. 1.307 26

 Guangzhou Zhongke Zhengchuan Environmental Protection Technology Co., Ltd. 1.307 28

 Inner Mongolia Dongshengmiao Mining Co., Ltd. 1.307 30

 China Non-ferrous Metal Mining (Group) Co., Ltd. 0.436 34

 Daye Non-ferrous Metals Group Holding Co., Ltd. 1.089 35

 Hunan Shizhuyuan Non-ferrous Metal Co., Ltd. 0.871 40

 Sinosteel Maanshan Mining Research Institute Co., Ltd. 0.871 41

 Hubei Liuguo Chemical Co., Ltd. 0.871 42

 Shanxi Kaixing Red Mud Development Co., Ltd. 0.871 43

 Huawei National Engineering Research Center for Efficient Recycling of Metal Mineral Resources Co., Ltd. 0.871 44

 Sinosteel Mining Development Co., Ltd. 0.871 45

 Sinosteel Hunan Phoenix Mining Co., Ltd. 0.871 46

 Bayannur Western Copper Co., Ltd. 0.871 47

 Hebei Ruisuo Solid Waste Engineering Technology Research Institute Co., Ltd. 0.436 48

 Daye non-ferrous metals company 0.871 49

Fig 6. Major inter-organisational patent cooperation.

We calculated the number of patents co-occurring among IPC classification numbers from the collected patent data containing 52 IPC classification numbers, and constructed the IPC co-occurrence matrix for analyzing the technology convergence network. The values of normalized degree, closeness, and betweenness of the constructed IPC co-occurrence matrix were calculated by using Ucinet software, as shown in Table 3. B03D103/02 (10.904) is the highest, which refers to "collectors," "depressants," and "ores" in the technology convergence network. This indicates that the three technology areas "collectors," "depressants," and "ores" are the most connected nodes in the technology convergence network, the most frequent fusion, and the highest degree of fusion, representing the core technology area of non-ferrous metal resource recycling. Closeness can determine the distance of nodes in the fusion network to disseminate information, and if the distance from nodes to other nodes is shorter, indicating a better mobility and faster fusion of technologies. C22B7/00 and B03B7/00, closest to the center degree, are relatively large, constituting the overall core of the network. However, they have a smaller center degree point and a larger intermediate center degree, which shows that although the sample organization is processing scrap and other materials for production, the device and method of processing waste refuse is not the core technology. Further, it constitutes the intermediate point of the network; moreover, it can promote the integration of other technologies. Betweenness indicates the possibility of technology convergence in the future and can be described as the potential influence of a technology, and if a technology possesses high mediating centrality, a mediating node with high potential to facilitate technology convergence in different domains is considered. Table 3 shows that C22B7/00 (4.092), B03B7/00 (3.804), and C22B7/04 (3.033) have the highest betweenness values. Although their normalized degree values are not high, B03B7/00 not only has a higher betweenness value, but also a larger closeness value. This means that non-ferrous metal resource recycling in China is "working-up raw materials other than ores (e.g., scrap, to produce non-ferrous metals or compounds thereof)," "combinations of wet processes or apparatus with other processes or apparatus (e.g., for dressing ores or garbage)," and "working up slag." They have a high potential for convergence in the areas of technology convergence networks, where they are directly or indirectly connected to other nodes, while facilitating the convergence of other technologies, and may be a key technology area for innovation in China in the future. In addition, we use Gephi software to visualize the IPC co-occurrence matrix. Fig 7 presents the network visualization. The thickness of the line represents the number of patents associated with the IPC node.

Table 3. IPC correlation and measurement metrics.

IPC class IPC subclass Description Normalized degree Closeness Betweenness 

B03D B03D101/02 Collectors 17.347 79.688 2.591

 B03D101/06 Depressants 12.465 75 1.832

 B03D103/02 Ores 10.904 73.913 1.806

 B03D1/018 Mixtures of inorganic and organic compounds 9.924 73.913 1.663

 B03D1/00 Flotation 9.744 80.952 2.794

 B03D101/04 Frother 9.804 76.119 1.93

 B03D1/002 Inorganic compounds 6.723 72.857 0.566

 B03D103/04 Non-sulfide ores 5.762 63.75 0.506

 B03D1/02 Inorganic compounds 5.142 66.234 0.566

 B03D1/012 Containing sulfur 5.122 68 0.812

 B03D1/008 Containing oxygen 3.882 64.557 0.632

 B03D1/08 Subsequent treatment of concentrated product 2.961 59.302 0.099

 B03D1/016 Macromolecular compounds 2.861 57.955 0.033

 B03D1/01 Containing nitrogen 2.021 66.234 0.946

 B03D1/004 Organic compounds 1.801 56.667 0.051

 B03D101/00 Specified effects produced by the flotation agents 1.140 57.303 0.079

 B03D1/014 Containing phosphorus 1.120 57.303 0.054

 B03D1/001 Flotation agents 0.88 57.955 0.201

B03B B03B7/00 Combinations of wet processes or apparatus with other processes or apparatus (e.g., for dressing ores or garbage) 9.024 80.952 3.804

 B03B1/00 Conditioning for facilitating separation by altering physical properties of the matter to be treated 7.143 75 1.967

 B03B9/00 General arrangement of separating plant (e.g., flow sheets) 6.323 73.913 2.035

 B03B1/04 By additives 3.161 64.557 0.696

 B03B9/06 Specially adapted for refuse 0.860 62.195 0.222

C22B C22B7/00 Working-up raw materials other than ores (e.g., scrap, to produce non-ferrous metals or compounds thereof) 2.821 75 4.092

 C22B15/00 Obtaining copper 2.721 76.119 2.495

 C22B1/02 Roasting processes (C22B1/16 takes precedence) 2.681 75 2.157

 C22B3/08 Sulfuric acid 1.721 68 1.242

 C22B7/04 Working-up slag 1.581 68 3.033

 C22B34/12 Obtaining titanium 1.341 58.621 0.93

 C22B1/00 Preliminary treatment of ores or scrap 1.16 67.105 1.913

 C22B34/22 Obtaining vanadium 1.18 64.557 2.629

 C22B26/10 Obtaining alkali metals 1.1 66.234 1.927

 C22B3/04 By leaching (C22B3/18 takes precedence) 0.98 65.385 0.927

 C22B59/00 Obtaining rare earth metals 0.96 60.714 0.926

 C22B11/00 Obtaining noble metals 0.88 62.195 0.455

 C22B19/30 From metallic residues or scraps 0.78 61.446 0.468

 C22B3/44 By chemical processes (C22B3/26, C22B3/42 take precedence) 0.84 62.963 0.617

 C22B23/00 Obtaining nickel or cobalt 0.8 62.195 0.585

 C22B21/00 Obtaining aluminum 0.76 57.303 0.235

 C22B13/00 Obtaining lead 0.6 59.302 0.303

 C22B1/24 Binding; Briquetting 0.62 62.195 0.694

 C22B11/08 Cyaniding 0.58 55.435 0.091

B03C B03C1/02 Acting directly on the substance being separated 1.821 70.833 2.076

 B03C1/015 By chemical treatment imparting magnetic properties to the material to be separated (e.g., roasting, reduction, and oxidation) 1.02 61.446 0.549

 B03C1/00 Magnetic separation 0.88 61.446 0.722

 B03C1/30 Combinations with other devices, not otherwise provided for 0.82 60.714 0.555

C21B C21B13/00 Making spongy iron or liquid steel by direct processes 1.02 60.714 1.340

 C21B3/06 Treatment of liquid slag 0.74 46.789 0.039

 C21B11/00 Making pig-iron other than in blast furnaces 0.58 50 0.058

B02C B02C21/00 Disintegrating plant with or without drying of the material (for grain B02C9/04) 0.66 60 0.299

C25C C25C1/12 Copper 0.62 57.955 0.207

C04B C04B7/147 Metallurgical slag 0.6 50 0.079

Fig 7. IPC association network diagram.

Concor analysis

To further analyze the technology convergence, we conducted a CONCOR calculation on the IPC association network using Ucinet. Fig 8 presents the results. The network modules are clustered into eight, and the clustering of each module indicates that the members have similar convergence. The first module is most closely connected to the second module and represents the maximum technological convergence, as a method for isolating useful and sustainable productive uses from waste materials. The second module is similar to the first module in that it is more convergent. The fifth module has fewer clustered nodes, but its intermediary role is obvious, as it connects the fourth module, which outputs the methodological and technological devices of waste treatment to the means of separating useful materials. Moreover, it is integrated with the sixth module, the refining of non-metallic materials, which serves a certain intermediary utility. The density matrix of the eight module was calculated using Ucinet (Table 4). Table 4 shows that Modules 1 and 2 have the highest correlation density, indicating that the elements in Module 1 are most closely linked to those in Module 2 in the technology convergence network. This reflects the high degree of convergence between the technology areas involved in Module 1 and those involved in Module 2. This indicates that China has a strong technological innovation capability in the fields of flotation, selective deposition method and separation of solid materials by liquid, wind shaker or wind jigger. They have formed very mature and stable fusion technologies in these two technology areas.

Fig 8. CONCOR analysis.

Table 4. IPC correlation module density matrix.

Cluster 1 2 3 4 5 6 7 8

1 21.800 11.678 2.818 1.636 0.519 0.409 0.318 0.073

2 11.678 10.400 1.909 1.295 0.338 0.341 0.136 0.055

3 2.818 1.909 1.000 0.500 3.857 3.000 1.250 0.500

4 1.636 1.295 0.500 0.667 0.214 0.156 0.063 0.400

5 0.519 0.338 3.857 0.214 4.048 1.375 0.786 2.543

6 0.409 0.341 3.000 0.156 1.375 2.107 0.656 0.600

7 0.318 0.136 1.250 0.063 0.786 0.656 6.333 2.700

8 0.073 0.055 0.500 0.400 2.543 0.600 2.700 3.200

R-squared = 0.371

QAP analysis

Regression Results

We use QAP to analyze the relationship between the constructed structural hole constraint index SHij, and Jaccard index matrix Jij, and then explore the influence of structural holes on technological convergence, based on patent cooperation networks. First, we use the 52 × 52 Jij matrix as the dependent variable for measuring the technology convergence network, in which the horizontal and vertical axes are permuted 2,000 times. The independent variables are SHij and (SH square)ij, the moderating variables are PCij and GDij, and the control variables are PSij and OSij. 

We perform QAP analysis on the above co-occurrence matrix using Ucinet software. We estimated a total of seven models in Table 7. Specifically, Model 1 includes only control variables. Models 2 and 3 examine the direct effect of structural holes on technology convergence. We construct a quadratic matrix of the structural hole constraint index (SH square)ij. Model 3 reveals a U-shaped curve relationship between the structural hole constraint index and technology convergence. Models 4, 5, and 6 show the moderating effect of the degree of patent cooperation and the distance of patent cooperation, as evidenced by the interaction effect with the quadratic terms of the structural hole constraint index and the structural hole constraint index. Model 7 is the full model with all variable matrices. 

First, Model 1 shows the regression equations for the control variables, followed by the regression equations for the independent variables (SHij and (SH square)ij) (Table 7, Model 2, 3). The regression coefficients of structural hole constraint index matrix (SHij) and quadratic structural hole constraint index matrix ((SH square)ij) are negative and positive and significant, respectively (β1=-0.068671, p<0.01, β2=0.068968, p<0.05). We found that the structural hole constraint index shows a positive U-shaped relationship with technology convergence, thus verifying H1. Hence, the more initial structural holes constrain the nodes in the network, the more unfavorable technology convergence is, and the more nodes occupy structural holes, thereby serving as intermediaries to promote the acquisition and absorption of knowledge and technology and effective technology moderation. As cooperation deepens, the organization crosses more structural holes, further increasing the risk cost, which is not as efficient as the cooperation sharing interdependence between nodes. This finding enriches the research on the impact of structural holes on innovation performance, Wen et.al argued [59] that the more structural holes that a firm crosses in an R&D network, the less constrained it is by the connected nodes, and the more complementary non-related diversified knowledge it can acquire, thus effectively absorbing and integrating internal and external knowledge and promoting the firm's exploitative and exploratory innovation outcomes. The innovation performance of the above studies is relatively homogeneous, and does not consider the linear relationship between structural holes and innovation performance. Instead, we examine the linear relationship between structural holes and technology convergence from the technology convergence perspective. We find a U-shaped relationship between the two. The less constrained an organization is in a patent cooperation network, the more technology it can absorb, the more nodes it can cross, and the more internal and external resources it can effectively access to promote technology convergence. After passing the inflection point, with the deepening of cooperation, the organizations' connection strengthens, and the ability of collaborative innovation increases. This proves that a joint cooperation can promote technology convergence better than individual enterprises' integration of knowledge and technology. At the same time, the resource-based theory found that the heterogeneous resources of organizations can help improve competitiveness. Our research presents the smaller the structural hole constraint in the patent cooperation network can absorb more external resources and integrate technologies. The results enrich the resource-based theory.

Second, the regression coefficients of the degree of patent cooperation (PCij), and the distance of cooperation (GDij) in Model 4, were positive and significant (β1=0.140354, p<0.01, β2=0.086126, p<0.01). This indicates that the closer the patent cooperation, the tighter the ties between organizations in the network, and the more beneficial for organizations to acquire new technologies and promote technology convergence. This is consistent with the findings of previous studies [21, 60, 61]. Meanwhile, the shorter the cooperation distance, the more favorable the flow of technology, which accelerates the frequency of cooperation and helps organizations accelerate technical uptake. However, the study of collaborative distance is controversial. Zhang and Tang argued [62] that the diversity of collaborative distance promotes innovation performance. And we found that the closer the cooperation distance is, the more it can promote technology convergence. The capability of Chinese non-ferrous metal resource recycling technology to innovate may be insufficient, and international cooperation may not prove to be enough. The non-ferrous metal output is highly polluting, at present, and can only be close to the raw material output to establish industrial parks and deal with the secondary use of waste. This makes long distance cooperation more difficult. Regarding H2, we find that the degree of patent cooperation in Model 5 positively moderates the positive U-shaped relationship between the structural hole constraint index ((SH square)ij×PCij) and technology convergence (β=0.061336, p<0.1). Fig 9 shows the moderating effects and the quadratic interaction coefficient between the structural hole constraint index and the degree of patent cooperation. From the figure, the structural hole constraint index shows a positive U-shaped relationship with technology convergence. When the degree of patent cooperation is higher, it strengthens the degree of interdependent ties in the network nodes, which induces the organizational structural hole constraint index to show a positive U-shaped non-linear relationship with technology convergence. Therefore, H2 is verified. This is similar to the findings of previous studies [63]. The greater the degree of cross border cooperation on innovation, the more it helps organizations to occupy a greater number of structural holes and promote innovation. Our study considers the moderating effect of the degree of patent cooperation, on the U-shaped relationship between structural holes and technology convergences. The study shows that the closer the patent cooperation, the higher will be the nodal structural holes in the cooperative network span, accessing more heterogeneous resources and accelerating technology convergence. In addition, the closer the organizations in the network cooperate, the more organizations become relatively less constrained by resources, which reduces the cost of accessing resources and information asymmetry. For example, in the field of non-ferrous metal resource recycling, China established a strategic alliance of technological innovation, which allowed the tenant group to improve its position in the cooperation network and integrate multiple technologies to form a complete industrial chain, including recycling-smelting-reproduction technologies. This result is attributable to the enterprise’s long-term industry-university-research cooperation with the Jiangsu Institute of Technology. Figure 9 shows that the bottom of the U-shaped curve shifts upward to the right when the degree of patent cooperation is greater. For example, in the field of non-ferrous metal resource recycling, China established a strategic alliance of technological innovation, which allowed the tenant group to improve its position in the cooperation network and integrate multiple technologies to form a complete industrial chain, including recycling-smelting-reproduction technologies. This result is attributable to the enterprise’s long-term industry-university-research cooperation with the Jiangsu Institute of Technology. Fig 9 shows that the bottom of the U-shaped curve shifts upward to the right, when the degree of patent cooperation is greater.

From Model 6, the cooperation distance negatively regulates the relationship between the structural hole constraint index and technology convergence (β=-0.030539, p<0.1). Fig 10 presents the moderating effect. From the Fig 10, the cooperation distance negatively regulates the relationship between the structural hole constraint index and technology convergence: the closer the cooperation distance, the smaller the structural hole constraint index, the less the node is constrained by external factors, and the more it can promote technology convergence. The farther the cooperation distance, the more the structural holes that must be crossed, and the more the intermediary bridges for the required knowledge technology. Therefore, H3 is verified. This is similar to the findings of a previous study [64]. Guan and Yan showed [26] that the geographical distance between two organizations in a patent collaboration network, has no moderating effect on innovation performance. Instead, we argue that the closer the collaboration distance, the less resource constrained the organization is, and the more quickly it can integrate internal and external technologies and promote technological convergence. It may be related to the fact that non-ferrous metal resource recycling technology field has specificity, non-ferrous metal waste needs to be recycled and treated nearby, and the proximity of cooperation between organizations is preferable, to reduce innovation costs and avoid secondary pollution. The previous research on resource-based theory ignored the effects on the use of resources. This study demonstrates that, the patent cooperation distance is an important context affecting the integration of technology. The reduced distance between the two organizations can prove conducive to mutual technical exchange, and technology convergence. This study further fills the gap of resource-based theoretical research.

Table 7. QAP regression results for technology convergence.

Jij Model 1 Model 2 Model 3 Model 4 Model 5 Model 6 Model 7

SHij 0.012863 -0.068671*** -0.082035*** -0.119666*** -0.067502*** -0.105121***

(SH square)ij 0.086618*** 0.068968** 0.064904** 0.066815** 0.066128** 0.068134**

PCij 0.141772*** 0.140354*** 0.125058*** 0.141314*** 0.126042***

GDij 0.086126*** 0.085750*** 0.091646*** 0.091266***

SHij×GDij -0.030539* -0.030481*

SHij×PCij -0.015443 -0.015551

(SH square)ij×PCij 0.061336* 0.061256*

PSij 0.204813*** 0.206259*** 0.210093*** 0.212847*** 0.212935*** 0.213801*** 0.213886***

OSij 0.000463 0.000816 0.000127 -0.000448 0.000181 -0.000581 0.000043

Intercept 0.000000 0.000000 0.000000 0.000000 0.000000 0.000000 0.000000

N 2652 2652 2652 2652 2652 2652 2652

R2 0.042 0.05 0.063 0.070 0.071 0.070 0.071

Adj.R2 0.042 0.049 0.061 0.068 0.068 0.068 0.068

Notes: * p < 0.1, ** p < 0.05, *** p<0.01.

Fig 9. The moderating effect of degree of patent cooperation network for this study.

Fig 10. The moderating effect of distance of patent cooperation network for this study."

Comment 5: Results: I suggest you discuss a little bit more the results of the case study with the previous (and recent) literature. I think you have space here to discuss your results and findings.

Response: Thank you for your constructive suggestion. Combined with your further comment, we expand our Discussion section. We have added some comparative findings from the literature analysis, which are marked in red in the Discussion section of the manuscript.

"Discussion

This study employed China’s invention patent data of non-ferrous metal resource recycling, to analyze the trend of technological convergence in 52 organizations with the co-occurrence matrices of cooperative patent applications. It examined the non-linear relationship between the structural hole constraint index and technological convergence, in the patent cooperation network, and employed social network theory to verify the moderating role of the degree and distance of patent cooperation. It has been previously noted in the literature [12,21,22] that R&D collaboration networks based on patented data, can facilitate technological convergence. However, less literature has considered the contextual factors influencing the impact of structural holes on technological convergence, which are important in influencing internal and external technological convergence, in conjunction with the collaborative network of green technology organizations in China. We fill this gap, and the main research findings are as follows.

This study makes several theoretical contributions to the previous literature. First, we contribute to the technology convergence literature by examining its dynamics for non-ferrous metal resource recycling in China. Previous literature has pointed out that technology convergence, as an important indicator of innovation performance, is also a powerful weapon for the outcome of internal and external technology convergence and for expanding market competition [35]. We found a lack of previous literature examining green technology convergence dynamics. Based on the social network analysis approach, we measure technology convergence mainly based on the IPC co-occurrence matrix. This study collects the non-ferrous metal resource recycling registrations in China based on the 52 types of primary IPC, involved in the non-ferrous metal resource recycling, where the value of two IPCs is the number of associated patents. Previous studies have only focused on the measurement of technology convergence in 3D printing [20], textile [35], and ICT [55] fields. In contrast, this study analyzed the Chinese non-ferrous metal resource recycling technology convergence, using normalized degree, closeness, and betweenness social network analysis indicators based on the construction of IPC co-occurrence matrix. This study found that the non-ferrous metal resource recycling in the core of the network are B03D101/02, B03D101/06, and B03D103/02. Given the intermediary position of C22B7/00 and B03B7/00, relative to other technology areas, the main cooperation advantage of China’s non-ferrous metal resources recycling technology is concentrated in the recycling of scrap metal. Equipment and technology methods for processing, scrap metal refining, and conversion and utilization are located at the fringes of the network. Thus, technology convergence proving insufficient, and organizations must improve the refining technology of scrap metal for effective moderation to other technology areas. This study may lead scholars to focus on the dynamics of technology convergence networks within the field of green technology convergence.

Second, this study proposes a new framework for exploring the drivers of technology convergence. We propose that the degree of structural hole constraints formed by inter-organizational patent cooperation affects technology convergence. We use QAP analysis to explore the relationship between structural hole and technology convergence, based on the construction of the structural hole constraint index matrix, and technology convergence matrix. We found that the structural hole constraint index shows a positive U-shaped effect on technology convergence. Organizations in cooperative networks occupy rich structural holes at the beginning of cooperation. They increasingly serve as intermediary bridges, conducive to technology cooperation and knowledge flow transfer. Our study extends the previous literature, and is consistent with the view that richer structural holes improve firm innovation performance [61]. However, previous literature has not examined the linear relationship between structural holes and technology convergence, and we argue that a moderate but not excessive number of structural holes in collaborative networks, would help organizations to better integrate technology. As cooperation deepens, organizations must cross more structural holes to obtain non-redundant resources and face increased risks and costs. Further, organizations increasingly rely on cooperative relationships, establish trusting cooperative relationships, conduct resource sharing, reduce the cost of searching for information, and effectively filter redundant information. As technological development increases in complexity and market requirements for technology efficacy increase, China’s non-ferrous resource recycling technology requires integrating multiple innovations such as recycling processing technology, recycling equipment, and waste refining. Long-term partnerships should be established between organizations to promote technological convergence. Meanwhile, the organization should shift its focus from the number of organizations involved in cooperation to the organizational structure in the industry chain. They should aim to allocate different R&D subjects in the upper, middle, and lower reaches of the technology chain to avoid generating excessive homogeneous technologies. Organizations should implement the industry-academia-research cooperation model and first consider the same type of organizational cooperation, conducive to the rapid transfer of knowledge and technology, given similar organizational structure, integration, and embedding in a larger cooperative network in small groups.

Third, the structural holes based on patent cooperation networks alone do not capture the full picture. The effect of structural holes on technology convergence may depend on two other contextual factors, namely the degree of patent cooperation and the distance of cooperation. We found that the degree of patent cooperation has a significant positive effect on the positive U-shaped relationship between the structural hole constraint index and technology convergence. Our findings enrich previous studies that innovation performance is jointly influenced by the contextual factors of structural capital and the degree of cooperation [63]. Thus, the smaller the constraint index in the patent cooperation network, the richer the structural capital, the cooperation can induce technology convergence. In addition, the cooperation distance should be emphasized in the innovation process and R&D cooperation. We found that the cooperation distance negatively regulates among the structural hole constraint index and technology convergence. The closer the cooperation distance effective the technology export to other members to acquire more resources, and induce more organizations to cooperate willingly. This is controversial with previous views [26,62]. As mentioned earlier, Guan and Yan’s [26] study has not been able to show that geographical distance of organizational cooperation plays a significant moderating role in organizational technological similarity affecting recombine innovation performance. Boschma and Frenken also argue [65] that geographical distance is not a necessary and sufficient requirement for innovation and knowledge sharing today. This differs from the conclusion reached in our study, where we speculate that non-ferrous metal resource recycling technological innovation is different from other industries such as new energy vehicles, artificial intelligence, and ICT, where the operational costs are higher and technological innovation is more expensive if the post use waste of non-ferrous metals is moved and transported to more distant plants and organizations for disposal. It is important for organizations to dispose of the related waste close to the site to save the cost and improve the efficiency of resource use, avoiding secondary pollution and huge waste of resources. In addition, the organizational cooperation system for related technologies in China is not mature, and the promotion of innovations and international cooperation needs to be improved. At present, it is basically a small group cooperating with each other in close proximity, which is conducive to the dissemination of knowledge and technology exchange, and can also apply and absorb new technologies to improve the refining technology after the recovery of non-ferrous metal waste, thus increasing the recyclable use of resources. Conversely, the geographical distance may bring synergistic effects and difficulties in knowledge transfer [66]. Therefore, it is necessary to absorb more organizations in the cooperative network to participate in cooperative innovation, strike a balance between independent innovation and cooperative innovation, and help organizations conduct technology convergence."

Reviewer #2

Thank you for your detailed comments, which have enlightened us to further strengthen our manuscript. We have carefully revised the manuscript according to your suggestions. Overall, we hope that our responses can meet your expectations.

Comment 1: The introduction section has been revised and shortened.

Response: Thank you for the helpful suggestion. Combined with this comment, we have rewritten the introduction more substantially. We have revised and trimmed redundant sections and focused on core concepts. Overall, the introduction section in our current manuscript is more concise and well laid out than the last manuscript.

Details of the revisions are marked red shown in the manuscript.

"Introduction

The recycling industry is a strategic emerging industry in China, which can significantly relieve the pressure on resources and the environment, thus promoting the construction of new urbanization and facilitating industrial restructuring [1]. Some scholars define the renewable resource industry as a series of activities regarding the recycling of renewable resources [2]. Non-ferrous metal recycling is an important renewable resource industry, and the market demand for metal materials and products is increasing. It is challenging for individual enterprises to develop effective technological innovation to recycle copper and lead. Furthermore, the technology is dated and requires the integration of complex technology [3-5]. Effective industry-academia-research cooperation to promote such integration with green technologies, such as resource recycling technology, can effectively enhance the development trend to break through the ‘neck’ of key common technologies, boost innovation capability, improve product quality, and achieve industrial restructuring and upgrading [6]. It is imperative to study effective cooperation between organizations to promote technology convergence.

Forming research and development (R&D) cooperative networks between organizations is an important way to implement technological innovation in high-tech industries [7]. Acquiring heterogeneous resources from collaborative networks is a vital function of network locations [8]. Granovetter [9] showed that the activities of organizations, which should not be limited to internal activities, should be extended for effective collaboration with other innovation organizations. When the uncertainty and ambiguity of technology deepens, organizations focus more on acquiring external resources for iterative updates [10]. Proximity cooperation enabled the organization to quickly cross the structural hole, break organizational boundaries to access diversified and heterogeneous technological resources, conduct effective technology convergence, and accelerate organizational innovation. Hence, proximity technology cooperation and structural holes can bring effective technology convergence. Thus, the mechanism of influence of structural holes and patent cooperation networks on technology convergence is worth studying [11].

However, prior studies show lesser engagement with China’s non-ferrous metal resource recycling technology. Furthermore, few studies explore the mechanism of structural holes’ influence on technology convergence. Therefore, this paper begins by describing the trends in the convergence of the analytical techniques for co-occurrence-based technology in non-ferrous metal recycling. Technology convergence can be divided into the following categories. The first category is based on whether the types of organizations cooperating in R&D projects are identical. If they belong to the same category, the technologies used may be similar, instead of two completely different technologies being integrated together. If the types of cooperating organizations belong to different categories, it is possible that two different types of technologies are integrated together [12]. In addition, some scholars have used a measurement approach to study technology convergence, based on academic literature and input-output (I/O) tables and research data [13-15]. The shortcomings of the aforementioned measurement techniques include difficulties in obtaining the limited data, a lag in the data, the requirement of a long period of observation, and the reflection of the integration of only industries and applications, without reflecting the integration of new technologies. However, the current popular measurement of technology convergence method is based on the co-occurrence of the International Patent Classification (IPC) number, for patented data, fused with social network analysis technology and patent citations [16-18]. In this study, the co-occurrence of patent data and IPC helps in constructing a patent cooperation network, to describe technology convergence of non-ferrous metal recycling based on social network theory. The value of co-occurrence of two IPC classification numbers, in a patent network, represents the number of patents in which these two IPCs occur simultaneously. Previous studies measured the Herfindahl index [19], entropy [20], and total number convergence patents [21] based on patent data, yet could not observe the process and degree of convergence of individual technology nodes in the overall network. The absolute number of patents also could not describe the dynamics of technology convergence, and different technological areas involved in convergence. However, a patent document with two or more IPC classification numbers implies that the patent involves multiple technologies, reflecting the source and development trend of the technologies and their applications [22]. This research constructs a patent network of IPC co-occurrence, based on the data from the patent information platform of key industries, of the State Intellectual Property Office of China. This can observe the composition of technologies, and the degree of integration of different technological nodes in the network. The patented information platform provides the information of non-ferrous metal recycling patents, primarily applied by innovative Chinese organizations. Describing the technological convergence dynamics proves beneficial for this study.

Hence, the second objective of this paper is to employ the social network theory and construct an inter-organizational R&D cooperation network, specifically divided into structural hole constraint index matrices. This analyses the effect of structural holes on technology convergence, under the contextual factors of patent cooperation degree and distance. Previous studies of R&D cooperation were based on the number of organized projects, knowledge output, technological diversity, and the number of employees involved in such R&D projects [12]. These are absolute data, not reflecting the flow of technology, the degree, and the specific organizational structure of cooperation. Faust and Wasserman [23] defines "one-mode" network, which speaks of some scholars measuring R&D cooperation with patent cooperation network. Patents are the R&D outputs that reflect the information and innovation outcomes of technology and its application, performances of collaborated R&D, and innovation paths [24]. Using the social network approach to construct a one-model patent cooperation network, this study observes the number of cooperated patents between two organizations in the network, reflecting the degree of R&D cooperation between them. The mobility of knowledge depends on the flexibility of the overall network, the nature of each nodal organization and the distance between them. However, this one-model social network cannot analyze the relationship between each network. For example, it cannot portray whether each nodal organization in the patent cooperation network plays the role of an intermediary or a core collaborator. It also cannot portray whether a structural hole (bridge) in the network, has any effect on the performance of innovation. At the same time, the advent of the era of big data and the development of the network integration processes have complicated the network formed by the relationship between individuals and groups. The general multiple regression analysis method cannot explore the non-independent relationships between different networks, and the covariance problem makes the ordinary least squares (OLS) method, based on time series and panel data, invalid. In this study, we try to use the quadratic assignment procedure (QAP) test to examine the hypothesis of the "relationship-relationship," and analyze the relationship data of the organization's patent cooperation. This method is based on the QAP test, which helps examine the relationship data under the organizational patent partnership and other factors affecting technological convergence, solving the auto-correlation problem, and producing relatively unbiased statistical results [25-27].

This reminder of the study is structured as follows. Section 2 presents the theoretical basis and research hypothesis. Section 3 presents the research design. Section 4 illustrates the results. Section 5 discusses the results and concludes the study. Fig 1 illustrates the theoretical framework model for this study."

Comment 2: The author should show the novelty of this research, the result, and the finding this research need requires discussion using relevant theories and previous research.

Response: Thank you for the helpful suggestion. Combined with your further suggestion, in the third paragraph of Regression Results of QAP Analysis, we add literature analysis to discuss our research results [59]. We found that R&D networks in which firms span more structural holes, are less constrained by connected nodes, and have access to more complementary unrelated diversified knowledge. This effectively assimilates and integrates internal and external knowledge, and promotes firms’ exploitative and exploratory innovation outcomes. However, the innovation performance of the above studies is relatively homogeneous, and does not consider the linear relationship between structural holes and innovation performance. Instead, we examine the linear relationship between the structural hole and technology convergence from another dimension of innovation performance, and draw richer conclusions. The resource-based theory found that the organizations acquire heterogeneous resources, in order to improve their competitiveness, and our research further enriches this theory.

In the fourth paragraph, we add literature [62] for comparative exploration, enriching the findings and interpreting the empirical results with specific cases. Regarding H2, we find that the degree of patent cooperation in Model 5 positively moderates the positive U-shaped relationship between the structural hole constraint index ((SH square)ij×PCij), and technology convergence (β=0.061336, p<0.1). The H2 is further verified in conjunction with Figure 9. 

In the fifth paragraph of this section, we added the literature [64] for discussion. It would be of interest to the readers that, our findings are exactly contrary to the literature [26], where Guan and Yan [26] show that the geographical distance between two organizations in a patent collaboration network has no moderating effect on innovation performance. Instead, we conclude that the closer the collaborative distance, the less resource-constrained the organization is and the more rapidly it can integrate internal and external technologies and promote technology convergence. "At the same time, we fill gaps existing in resource-based theoretical research. The research on such theories, has not considered the context of resources on enterprise performance. In this study, we point that the distance of organizing patent cooperation is an important situational factor affecting technological convergence. 

Thus, in the revised Discussion section, we highlight the novelty of the study. Details of the revisions are marked red shown in the manuscript. 

"Empirical results and analysis

QAP analysis

 Regression results

We use QAP to analyze the relationship between the constructed structural hole constraint index SHij, and Jaccard index matrix Jij, and then explore the influence of structural holes on technological convergence, based on patent cooperation networks. First, we use the 52 × 52 Jij matrix as the dependent variable for measuring the technology convergence network, in which the horizontal and vertical axes are permuted 2,000 times. The independent variables are SHij and (SH square)ij, the moderating variables are PCij and GDij, and the control variables are PSij and OSij. We perform QAP analysis on the above co-occurrence matrix using Ucinet software.

We estimated a total of seven models in Table 7. Specifically, Model 1 includes only control variables. Models 2 and 3 examine the direct effect of structural holes on technology convergence. We construct a quadratic matrix of the structural hole constraint index (SH square)ij. Model 3 reveals a U-shaped curve relationship between the structural hole constraint index and technology convergence. Models 4, 5, and 6 show the moderating effect of the degree of patent cooperation and the distance of patent cooperation, as evidenced by the interaction effect with the quadratic terms of the structural hole constraint index and the structural hole constraint index. Model 7 is the full model with all variable matrices. 

First, Model 1 shows the regression equations for the control variables, followed by the regression equations for the independent variables (SHij and (SH square)ij) (Table 7, Model 2, 3). The regression coefficients of structural hole constraint index matrix (SHij) and quadratic structural hole constraint index matrix ((SH square)ij) are negative and positive and significant, respectively (β1=-0.068671, p<0.01, β2=0.068968, p<0.05). We found that the structural hole constraint index shows a positive U-shaped relationship with technology convergence, thus verifying H1. Hence, the more initial structural holes constrain the nodes in the network, the more unfavorable technology convergence is, and the more nodes occupy structural holes, thereby serving as intermediaries to promote the acquisition and absorption of knowledge and technology and effective technology moderation. As cooperation deepens, the organization crosses more structural holes, further increasing the risk cost, which is not as efficient as the cooperation sharing interdependence between nodes. This finding enriches the research on the impact of structural holes on innovation performance, Wen et.al argued [59] that the more structural holes that a firm crosses in an R&D network, the less constrained it is by the connected nodes, and the more complementary non-related diversified knowledge it can acquire, thus effectively absorbing and integrating internal and external knowledge and promoting the firm's exploitative and exploratory innovation outcomes. The innovation performance of the above studies is relatively homogeneous, and does not consider the linear relationship between structural holes and innovation performance. Instead, we examine the linear relationship between structural holes and technology convergence from the technology convergence perspective. We find a U-shaped relationship between the two. The less constrained an organization is in a patent cooperation network, the more technology it can absorb, the more nodes it can cross, and the more internal and external resources it can effectively access to promote technology convergence. After passing the inflection point, with the deepening of cooperation, the organizations' connection strengthens, and the ability of collaborative innovation increases. This proves that a joint cooperation can promote technology convergence better than individual enterprises' integration of knowledge and technology. At the same time, the resource-based theory found that the heterogeneous resources of organizations can help improve competitiveness. Our research presents the smaller the structural hole constraint in the patent cooperation network can absorb more external resources and integrate technologies. The results enrich the resource-based theory.

Second, the regression coefficients of the degree of patent cooperation (PCij), and the distance of cooperation (GDij) in Model 4, were positive and significant (β1=0.140354, p<0.01, β2=0.086126, p<0.01). This indicates that the closer the patent cooperation, the tighter the ties between organizations in the network, and the more beneficial for organizations to acquire new technologies and promote technology convergence. This is consistent with the findings of previous studies [21, 60, 61]. Meanwhile, the shorter the cooperation distance, the more favorable the flow of technology, which accelerates the frequency of cooperation and helps organizations accelerate technical uptake. However, the study of collaborative distance is controversial. Zhang and Tang argued [62] that the diversity of collaborative distance promotes innovation performance. And we found that the closer the cooperation distance is, the more it can promote technology convergence. The capability of Chinese non-ferrous metal resource recycling technology to innovate may be insufficient, and international cooperation may not prove to be enough. The non-ferrous metal output is highly polluting, at present, and can only be close to the raw material output to establish industrial parks and deal with the secondary use of waste. This makes long distance cooperation more difficult. Regarding H2, we find that the degree of patent cooperation in Model 5 positively moderates the positive U-shaped relationship between the structural hole constraint index ((SH square)ij×PCij) and technology convergence (β=0.061336, p<0.1). Fig 9 shows the moderating effects and the quadratic interaction coefficient between the structural hole constraint index and the degree of patent cooperation. From the figure, the structural hole constraint index shows a positive U-shaped relationship with technology convergence. When the degree of patent cooperation is higher, it strengthens the degree of interdependent ties in the network nodes, which induces the organizational structural hole constraint index to show a positive U-shaped non-linear relationship with technology convergence. Therefore, H2 is verified. This is similar to the findings of previous studies [63]. The greater the degree of cross border cooperation on innovation, the more it helps organizations to occupy a greater number of structural holes and promote innovation. Our study considers the moderating effect of the degree of patent cooperation, on the U-shaped relationship between structural holes and technology convergences. The study shows that the closer the patent cooperation, the higher will be the nodal structural holes in the cooperative network span, accessing more heterogeneous resources and accelerating technology convergence. In addition, the closer the organizations in the network cooperate, the more organizations become relatively less constrained by resources, which reduces the cost of accessing resources and information asymmetry. For example, in the field of non-ferrous metal resource recycling, China established a strategic alliance of technological innovation, which allowed the tenant group to improve its position in the cooperation network and integrate multiple technologies to form a complete industrial chain, including recycling-smelting-reproduction technologies. This result is attributable to the enterprise’s long-term industry-university-research cooperation with the Jiangsu Institute of Technology. Figure 9 shows that the bottom of the U-shaped curve shifts upward to the right when the degree of patent cooperation is greater. For example, in the field of non-ferrous metal resource recycling, China established a strategic alliance of technological innovation, which allowed the tenant group to improve its position in the cooperation network and integrate multiple technologies to form a complete industrial chain, including recycling-smelting-reproduction technologies. This result is attributable to the enterprise’s long-term industry-university-research cooperation with the Jiangsu Institute of Technology. Fig 9 shows that the bottom of the U-shaped curve shifts upward to the right, when the degree of patent cooperation is greater.

From Model 6, the cooperation distance negatively regulates the relationship between the structural hole constraint index and technology convergence (β=-0.030539, p<0.1). Fig 10 presents the moderating effect. From the Fig 10, the cooperation distance negatively regulates the relationship between the structural hole constraint index and technology convergence: the closer the cooperation distance, the smaller the structural hole constraint index, the less the node is constrained by external factors, and the more it can promote technology convergence. The farther the cooperation distance, the more the structural holes that must be crossed, and the more the intermediary bridges for the required knowledge technology. Therefore, H3 is verified. This is similar to the findings of a previous study [64]. Guan and Yan showed [26] that the geographical distance between two organizations in a patent collaboration network, has no moderating effect on innovation performance. Instead, we argue that the closer the collaboration distance, the less resource constrained the organization is, and the more quickly it can integrate internal and external technologies and promote technological convergence. It may be related to the fact that non-ferrous metal resource recycling technology field has specificity, non-ferrous metal waste needs to be recycled and treated nearby, and the proximity of cooperation between organizations is preferable, to reduce innovation costs and avoid secondary pollution. The previous research on resource-based theory ignored the effects on the use of resources. This study demonstrates that, the patent cooperation distance is an important context affecting the integration of technology. The reduced distance between the two organizations can prove conducive to mutual technical exchange, and technology convergence. This study further fills the gap of resource-based theoretical research.

Table 7. QAP regression results for technology convergence.

Jij Model 1 Model 2 Model 3 Model 4 Model 5 Model 6 Model 7

SHij 0.012863 -0.068671*** -0.082035*** -0.119666*** -0.067502*** -0.105121***

(SH square)ij 0.086618*** 0.068968** 0.064904** 0.066815** 0.066128** 0.068134**

PCij 0.141772*** 0.140354*** 0.125058*** 0.141314*** 0.126042***

GDij 0.086126*** 0.085750*** 0.091646*** 0.091266***

SHij×GDij -0.030539* -0.030481*

SHij×PCij -0.015443 -0.015551

(SH square)ij×PCij 0.061336* 0.061256*

PSij 0.204813*** 0.206259*** 0.210093*** 0.212847*** 0.212935*** 0.213801*** 0.213886***

OSij 0.000463 0.000816 0.000127 -0.000448 0.000181 -0.000581 0.000043

Intercept 0.000000 0.000000 0.000000 0.000000 0.000000 0.000000 0.000000

N 2652 2652 2652 2652 2652 2652 2652

R2 0.042 0.05 0.063 0.070 0.071 0.070 0.071

Adj.R2 0.042 0.049 0.061 0.068 0.068 0.068 0.068

Notes: * p < 0.1, ** p < 0.05, *** p<0.01.

Fig 9. The moderating effect of degree of patent cooperation network for this study.

Fig 10. The moderating effect of distance of patent cooperation network for this study.

Discussion

This study employed China’s invention patent data of non-ferrous metal resource recycling, to analyze the trend of technological convergence in 52 organizations with the co-occurrence matrices of cooperative patent applications. It examined the non-linear relationship between the structural hole constraint index and technological convergence, in the patent cooperation network, and employed social network theory to verify the moderating role of the degree and distance of patent cooperation. It has been previously noted in the literature [12,21,22] that R&D collaboration networks based on patented data, can facilitate technological convergence. However, less literature has considered the contextual factors influencing the impact of structural holes on technological convergence, which are important in influencing internal and external technological convergence, in conjunction with the collaborative network of green technology organizations in China. We fill this gap, and the main research findings are as follows."

Comment 3: Please show what is the contribution of the research to the parties who need these results in accordance with the objectives of this research.

Response: Thank you for your constructive suggestion. Combined with your further suggestion in Comment 3, we emphasized the contribution of our study in accordance with the objective of this research in the Conclusion and Insights section. The results and findings of our study could be offered to both managers and technological policymakers. The details can be found in the red marked sections for Conclusion and Insights.

"Finally, this study provides insights to managers and technological policymakers. For managers, the results of the study provide theoretical support, for strategizing about innovation and partner selection of organizations in R&D collaboration networks, thereby promoting technological innovation capabilities. When collaborating on patents, this study shows that universities, leading the cooperation with more structural capital, have more patents in non-ferrous metal resource recycling. Thus, managers should encourage industry-academia-research or patent alliances. Further, same- (different-) organization cooperation is more (less) conducive to technological convergence capabilities, which is consistent with prior findings about a Korean ICT firm cooperation being more conducive to technological convergence than firm-university cooperation [67]. When seeking partners, managers should prioritize cooperation between geographically close organizations and inter-organizational environment, to expand its scope and access to resources. Through initial small cooperation, they should absorb organizations with rich structural holes into the network, balance independent and cooperative innovations, and form a competitive advantage in a larger network. For technological policymakers, they should designate policies to promote cooperation, with universities in the leading role to form a complete cooperation chain, cultivate core innovation subjects, and increase structural capital. The technological policymakers should encourage organizations to build innovation ecosystems. In lieu of non-ferrous metal resource recycling in China, government agencies should support monopolies, and enterprises with strong technological innovation capabilities, as organizations that span more structural holes in R&D cooperation networks, and promote good cooperation with other heterogeneous organizations (especially universities). The technological policymakers need to guide organizations toward cross-border collaboration, to develop key technologies for resource recycling, and to provide the necessary funding or preferential policies. In addition, the government needs to encourage organizations to collaborate with firms in unrelated technological fields, to explore new technologies and conduct effective technological convergence."

Comment 4: Lastly, the discussion section is somewhat weak. The authors should clearly state the key lessons learned. There is a need to strengthen the argument of the paper, most of your assertions are loosely accompanied by excessive juxtaposing. Please address this issue.

Response: Thank you for your important comments, which have improved our study. Combined with your further suggestion in Comment 4, we have enhanced the discussion of the manuscript, to clearly state the key lessons learned from this study. Moreover, to strengthen the argument of the paper, we justify our points based on findings from previous literature and we elaborate the Discussion section.

For example, in the first paragraph of Discussion, we added literature [12,21,22] and summarized it, highlighting the theoretical and practical contributions and innovations of our study.

In the second paragraph, we argued that the study has several theoretical contributions to the previous literature. We summarize our research work and illustrate its significance and innovations, through several important literature [20,35,55], that speaks of previous studies neglecting green technologies. Instead, our work lies in the construction of an IPC co-occurrence matrix based on the social network analysis indicators like degree of normalization, closeness, and betweenness, that analyze the non-ferrous metal resource recycling in China. Further, the core technology and the future technical trends are analyzed.

In the third paragraph, we present a detailed theoretical summary of this study along with comparative literature, discussing the proposal of a new framework for exploring the drivers of technological convergence. We begin with a comparative analysis of the literature [61]. Our findings not only reinforce previous perspectives but also compensate for the research related to the linear relationship between research structural holes and technological convergence. Based on this analysis, we provide an in-depth discussion of our findings and make relevant recommendations.

In the fourth paragraph, we argue that our findings enrich previous studies showcasing that innovation performance is jointly influenced by contextual factors of structural capital and degree of cooperation. We analyze this through an in-depth discussion with the research perspectives of the literature [26,62,63,65,66], where we argue that the cooperation distance negatively regulates the structural hole constraint index and technological convergence. The closer the cooperation distance, the more effective the technical export to other members for acquiring more resources, and inducing more organizations to cooperate willingly. This is contrary to the view of the literature [26,62]. We have discussed and explained the case in a reasonable way through the actual situation, and made relevant suggestions.

Finally, we adjusted the statements in the Discussion section, modified the sentence structure, and removed some redundant and unnecessary phrases, to make the expository analysis more concise and succinct, highlighting the conclusions and contributions of the study, making it clear to the reader at a glance.

We have added more content to the Discussion section and highlighted these newly added portions in red.

"Discussion

This study employed China’s invention patent data of non-ferrous metal resource recycling, to analyze the trend of technological convergence in 52 organizations with the co-occurrence matrices of cooperative patent applications. It examined the non-linear relationship between the structural hole constraint index and technological convergence, in the patent cooperation network, and employed social network theory to verify the moderating role of the degree and distance of patent cooperation. It has been previously noted in the literature [12,21,22] that R&D collaboration networks based on patented data, can facilitate technological convergence. However, less literature has considered the contextual factors influencing the impact of structural holes on technological convergence, which are important in influencing internal and external technological convergence, in conjunction with the collaborative network of green technology organizations in China. We fill this gap, and the main research findings are as follows.

This study makes several theoretical contributions to the previous literature. First, we contribute to the technology convergence literature by examining its dynamics for non-ferrous metal resource recycling in China. Previous literature has pointed out that technology convergence, as an important indicator of innovation performance, is also a powerful weapon for the outcome of internal and external technology convergence and for expanding market competition [35]. We found a lack of previous literature examining green technology convergence dynamics. Based on the social network analysis approach, we measure technology convergence mainly based on the IPC co-occurrence matrix. This study collects the non-ferrous metal resource recycling registrations in China based on the 52 types of primary IPC, involved in the non-ferrous metal resource recycling, where the value of two IPCs is the number of associated patents. Previous studies have only focused on the measurement of technology convergence in 3D printing [20], textile [35], and ICT [55] fields. In contrast, this study analyzed the Chinese non-ferrous metal resource recycling technology convergence, using normalized degree, closeness, and betweenness social network analysis indicators based on the construction of IPC co-occurrence matrix. This study found that the non-ferrous metal resource recycling in the core of the network are B03D101/02, B03D101/06, and B03D103/02. Given the intermediary position of C22B7/00 and B03B7/00, relative to other technology areas, the main cooperation advantage of China’s non-ferrous metal resources recycling technology is concentrated in the recycling of scrap metal. Equipment and technology methods for processing, scrap metal refining, and conversion and utilization are located at the fringes of the network. Thus, technology convergence proving insufficient, and organizations must improve the refining technology of scrap metal for effective moderation to other technology areas. This study may lead scholars to focus on the dynamics of technology convergence networks within the field of green technology convergence.

Second, this study proposes a new framework for exploring the drivers of technology convergence. We propose that the degree of structural hole constraints formed by inter-organizational patent cooperation affects technology convergence. We use QAP analysis to explore the relationship between structural hole and technology convergence, based on the construction of the structural hole constraint index matrix, and technology convergence matrix. We found that the structural hole constraint index shows a positive U-shaped effect on technology convergence. Organizations in cooperative networks occupy rich structural holes at the beginning of cooperation. They increasingly serve as intermediary bridges, conducive to technology cooperation and knowledge flow transfer. Our study extends the previous literature, and is consistent with the view that richer structural holes improve firm innovation performance [61]. However, previous literature has not examined the linear relationship between structural holes and technology convergence, and we argue that a moderate but not excessive number of structural holes in collaborative networks, would help organizations to better integrate technology. As cooperation deepens, organizations must cross more structural holes to obtain non-redundant resources and face increased risks and costs. Further, organizations increasingly rely on cooperative relationships, establish trusting cooperative relationships, conduct resource sharing, reduce the cost of searching for information, and effectively filter redundant information. As technological development increases in complexity and market requirements for technology efficacy increase, China’s non-ferrous resource recycling technology requires integrating multiple innovations such as recycling processing technology, recycling equipment, and waste refining. Long-term partnerships should be established between organizations to promote technological convergence. Meanwhile, the organization should shift its focus from the number of organizations involved in cooperation to the organizational structure in the industry chain. They should aim to allocate different R&D subjects in the upper, middle, and lower reaches of the technology chain to avoid generating excessive homogeneous technologies. Organizations should implement the industry-academia-research cooperation model and first consider the same type of organizational cooperation, conducive to the rapid transfer of knowledge and technology, given similar organizational structure, integration, and embedding in a larger cooperative network in small groups.

Third, the structural holes based on patent cooperation networks alone do not capture the full picture. The effect of structural holes on technology convergence may depend on two other contextual factors, namely the degree of patent cooperation and the distance of cooperation. We found that the degree of patent cooperation has a significant positive effect on the positive U-shaped relationship between the structural hole constraint index and technology convergence. Our findings enrich previous studies that innovation performance is jointly influenced by the contextual factors of structural capital and the degree of cooperation [63]. Thus, the smaller the constraint index in the patent cooperation network, the richer the structural capital, the cooperation can induce technology convergence. In addition, the cooperation distance should be emphasized in the innovation process and R&D cooperation. We found that the cooperation distance negatively regulates among the structural hole constraint index and technology convergence. The closer the cooperation distance effective the technology export to other members to acquire more resources, and induce more organizations to cooperate willingly. This is controversial with previous views [26,62]. As mentioned earlier, Guan and Yan’s [26] study has not been able to show that geographical distance of organizational cooperation plays a significant moderating role in organizational technological similarity affecting recombine innovation performance. Boschma and Frenken also argue [65] that geographical distance is not a necessary and sufficient requirement for innovation and knowledge sharing today. This differs from the conclusion reached in our study, where we speculate that non-ferrous metal resource recycling technological innovation is different from other industries such as new energy vehicles, artificial intelligence, and ICT, where the operational costs are higher and technological innovation is more expensive if the post use waste of non-ferrous metals is moved and transported to more distant plants and organizations for disposal. It is important for organizations to dispose of the related waste close to the site to save the cost and improve the efficiency of resource use, avoiding secondary pollution and huge waste of resources. In addition, the organizational cooperation system for related technologies in China is not mature, and the promotion of innovations and international cooperation needs to be improved. At present, it is basically a small group cooperating with each other in close proximity, which is conducive to the dissemination of knowledge and technology exchange, and can also apply and absorb new technologies to improve the refining technology after the recovery of non-ferrous metal waste, thus increasing the recyclable use of resources. Conversely, the geographical distance may bring synergistic effects and difficulties in knowledge transfer [66]. Therefore, it is necessary to absorb more organizations in the cooperative network to participate in cooperative innovation, strike a balance between independent innovation and cooperative innovation, and help organizations conduct technology convergence."

---

## [Decision Letter · Decision Letter 1]

6 Jul 2022

China's non-ferrous metal recycling technology convergence and driving factors: a quadratic assignment procedure analysis based on patent collaboration-based network structural hole

PONE-D-22-11322R1

Dear Dr. Zor,

We’re pleased to inform you that your manuscript has been judged scientifically suitable for publication and will be formally accepted for publication once it meets all outstanding technical requirements.

Kind regards,

Ahmed Mancy Mosa, Ph.D.

Academic Editor

PLOS ONE

Additional Editor Comments (optional):

Reviewers' comments:

Reviewer's Responses to Questions

**Comments to the Author**

1. If the authors have adequately addressed your comments raised in a previous round of review and you feel that this manuscript is now acceptable for publication, you may indicate that here to bypass the “Comments to the Author” section, enter your conflict of interest statement in the “Confidential to Editor” section, and submit your "Accept" recommendation.

Reviewer #1: All comments have been addressed

Reviewer #2: (No Response)

2. Is the manuscript technically sound, and do the data support the conclusions?

Reviewer #1: Yes

Reviewer #2: (No Response)

3. Has the statistical analysis been performed appropriately and rigorously? 

Reviewer #1: Yes

Reviewer #2: (No Response)

4. Have the authors made all data underlying the findings in their manuscript fully available?

Reviewer #1: Yes

Reviewer #2: (No Response)

5. Is the manuscript presented in an intelligible fashion and written in standard English?

Reviewer #1: Yes

Reviewer #2: (No Response)

6. Review Comments to the Author

Reviewer #1: Dear author/s,

Thank you for your efforts in revising the manuscript. I believe you did a great job in the revision and the changes alleviated my concerns regarding the manuscript. Therefore, I recommend its publication.

Reviewer #2: (No Response)

7. PLOS authors have the option to publish the peer review history of their article (what does this mean?). If published, this will include your full peer review and any attached files.

Reviewer #1: No

Reviewer #2: No

---

## [Editor Report · Acceptance letter]

14 Jul 2022

PONE-D-22-11322R1 

China's non-ferrous metal recycling technology convergence and driving factors: a quadratic assignment procedure analysis based on patent collaboration-based network structural hole 

Dear Dr. Zor:

I'm pleased to inform you that your manuscript has been deemed suitable for publication in PLOS ONE. Congratulations! Your manuscript is now with our production department. 

Kind regards, 

on behalf of

Dr. Ahmed Mancy Mosa 

Academic Editor

PLOS ONE